# Pyqcm: An open-source Python library for quantum cluster methods

Théo N. Dionne, Alexandre Foley, Moïse Rousseau, David Sénéchal[*]

Département de physique and Institut quantique, Université de Sherbrooke, Sherbrooke, Québec, Canada J1K 2R1 * david.senechal@usherbrooke.ca

May 31, 2023

## Abstract

**Pyqcm is a Python/C++ library that implements a few quantum cluster methods with an exact diagonalization impurity solver. Quantum cluster methods are used in the study of strongly correlated electrons to provide an approximate solution to Hubbard-like models. The methods covered by this library are Cluster Perturbation Theory (CPT), the Variational Cluster Approach (VCA) and Cellular (or Cluster) Dynamical Mean Field Theory (CDMFT). The impurity solver (the technique used to compute the cluster's interacting Green function) is exact diagonalization from sparse matrices, using the Lanczos algorithm and variants thereof. The core library is written in C++ for performance, but the interface is in Python, for ease of use and interoperability with the numerical Python ecosystem. The library is distributed under the GPL license.**

## Contents

arXiv:2305.18643v1 [cond-mat.str-el] 29 May 2023

# 1   Introduction

Our understanding of the solid state has long been based on simple paradigms: metals can be understood in terms of quasi-independent electrons, undergoing occasional collisions; at the other extreme, magnets are understood in terms of the spins of localized electrons. But between these paradigms lies a spectrum of materials that defy comprehension in terms of these simple pictures, even though they may show characteristics of both. High-temperature superconductors are the prototype of such *strongly correlated quantum materials*. Such materials can display a variety of fascinating properties, from superconductivity to exotic magnetism, charge ordering, transitions between insulating and conducting behavior, spontaneous violation of time-reversal symmetry, etc.

Strongly correlated behavior is very often described theoretically using the Hubbard model, including variations thereof involving more than one band, extended interactions, and so on. Hubbard-like models are notoriously difficult to deal with. In the last 35 years or so, many computational methods were devised or significantly improved in order to treat such models. Most notorious is dynamical mean field theory (DMFT) [1, 2], which led to new insights into the Mott metal-insulator transition. A key approximation within DMFT is that the system's self-energy $\Sigma$ is momentum-independent, and depends only on frequency. To improve on this, *quantum cluster methods* (QCM) have been proposed,

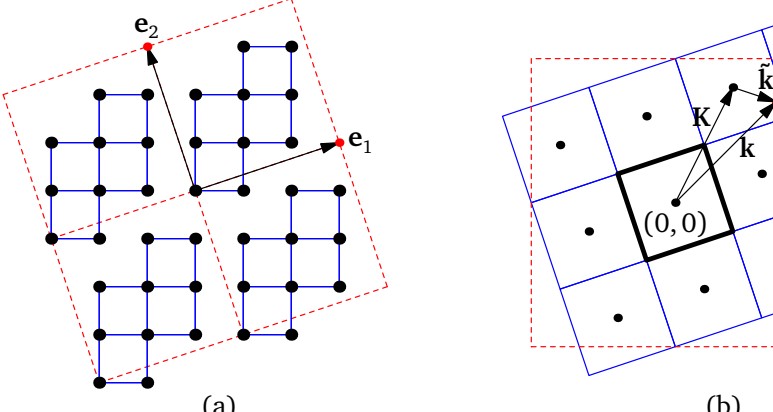

Figure 1: (a) A 10-site cluster and the corresponding super-lattice vectors. (b) The associated reduced Brillouin $BZ_\Gamma$ (thick black square); a wave-vector $\mathbf{k}$ has a unique decomposition $\mathbf{k} = \tilde{\mathbf{k}} + \mathbf{K}$, where $\mathbf{K}$ is one of the $L$ elements of the reciprocal super-lattice that belongs to the original Brillouin zone $BZ_\gamma$ (red dashed square).

in which the momentum dependence of the self-energy is not completely neglected, but restricted to a few points (or patches) in the Brillouin zone (for a review, see, e.g., [3, 4]). In the spatial domain, this amounts to including non-local components in the self-energy within a small cluster of atomic sites or orbitals. Such quantum cluster methods include cluster perturbation theory (CPT) [5,6], the cellular dynamical mean-field theory (CDMFT) [7], the dynamical cluster approximation (DCA) [8,9] and the variational cluster approach (VCA) [10].

Here we presents `pyqcm`, an open-source library for CPT, CDMFT and VCA based on an exact-diagonalization (ED) solver. This library has been developed over 20 years, but has only been given a Python interface in the last 4 years. This gave it more flexibility and ease of use, which justifies its public release. The first sections of this paper constitute a review of the different quantum cluster methods covered in the library. In the last section we will describe the overall architecture of the library and provide simple examples of its use, many more examples being available in the library's distribution.

Let us start by writing the Hamiltonian of the one-band Hubbard model, mostly to set the notation:

$$H = \sum_{\mathbf{r},\mathbf{r}',\sigma} t_{\mathbf{r}\mathbf{r}'} c^\dagger_{\mathbf{r}\sigma} c_{\mathbf{r}'\sigma} + U \sum_i n_{\mathbf{r}\uparrow} n_{\mathbf{r}\downarrow} - \mu \sum_{\mathbf{r}} n_{\mathbf{r}} \tag{1}$$

Here $\mathbf{r}$ denotes a site of a Bravais lattice $\gamma$, $c_{\mathbf{r}'\sigma}$ is the annihilation operator of an electron of spin $\sigma$ in a Wannier state centered at lattice site $\mathbf{r}$, $t_{\mathbf{r}\mathbf{r}'}$ is the hopping amplitude between Wannier states located at sites $\mathbf{r}$ and $\mathbf{r}'$, $U$ is the one-site Coulomb repulsion and $\mu$ is the chemical potential, which we find convenient to include in the Hamiltonian. We may assume, for counting purposes, that the lattice $\gamma$ is periodic, with a large (i.e., billions) but finite number of sites $N$. Multi-band Hubbard models are a simple extension of this, that we will introduce later as needed.

## 2 Clusters and super-lattices

Cluster methods are based on a tiling of the original lattice $\gamma$ with identical clusters of $L$ sites each. Mathematically, this corresponds to introducing a super-lattice $\Gamma$, whose

sites, labeled by vector with tildes ($\tilde{\mathbf{r}}$, $\tilde{\mathbf{r}}'$, etc), form a subset of the lattice $\gamma$. Every site $\tilde{\mathbf{r}}$ of the super-lattice may be expressed as an integer linear combination of $D$ basis vectors $\mathbf{e}_1, \ldots, \mathbf{e}_D$ belonging to $\gamma$. Associated with each site of $\Gamma$ is a cluster of $L$ sites, whose shape is not uniquely determined by the super-lattice structure. The sites within the clusters will be labeled by their vector position (in capitals): $\mathbf{R}$, $\mathbf{R}'$, etc. Each position $\mathbf{r}$ of the original lattice $\gamma$ can thus be uniquely expressed as a combination of a super-lattice vector $\tilde{\mathbf{r}}$ and of a position $\mathbf{R}$ within the cluster: $\mathbf{r} = \tilde{\mathbf{r}} + \mathbf{R}$ (see Fig. 1(a)).

The number of sites in the cluster is simply the ratio of the unit cell volumes of the two lattices. In $D = 3$, this is

$$L = \frac{V_\Gamma}{V_\gamma} = |(\mathbf{e}_1 \wedge \mathbf{e}_2) \cdot \mathbf{e}_3| \tag{2}$$

(the above formulae can be adapted to $D = 2$ by setting $\mathbf{e}_3 = (0, 0, 1)$).

The Brillouin zone of the original lattice, denoted $\mathrm{BZ}_\gamma$, contains $L$ points belonging to the reciprocal super-lattice $\Gamma^*$. Correspondingly, the Brillouin zone of the super-lattice, $\mathrm{BZ}_\Gamma$, is $L$ times smaller than the original Brillouin zone. Any wave-vector $\mathbf{k}$ of the original Brillouin zone can be uniquely expressed as

$$\mathbf{k} = \mathbf{K} + \tilde{\mathbf{k}} , \tag{3}$$

where $\mathbf{K}$ belongs both to the reciprocal super-lattice *and* to $\mathrm{BZ}_\gamma$, and $\tilde{\mathbf{k}}$ belongs to $\mathrm{BZ}_\Gamma$ (see Fig. 1(b)).

## 2.1  Partial Fourier transforms

The passage between momentum space and real space, by discrete Fourier transforms, can be done either directly ($\mathbf{r} \leftrightarrow \mathbf{k}$), or independently for cluster and super-lattice sites ($\tilde{\mathbf{r}} \leftrightarrow \tilde{\mathbf{k}}$ and $\mathbf{R} \leftrightarrow \mathbf{Q}$). This can be encoded into unitary matrices $\mathbf{U}^\gamma$, $\mathbf{U}^\Gamma$ and $\mathbf{U}^c$ defined as follows:

$$U_{\mathbf{k},\mathbf{r}}^\gamma = \frac{1}{\sqrt{N}} \mathrm{e}^{-i\mathbf{k}\cdot\mathbf{r}} \qquad U_{\tilde{\mathbf{k}},\tilde{\mathbf{r}}}^\Gamma = \sqrt{\frac{L}{N}} \mathrm{e}^{-i\tilde{\mathbf{k}}\cdot\tilde{\mathbf{r}}} \qquad U_{\mathbf{K},\mathbf{R}}^c = \frac{1}{\sqrt{L}} \mathrm{e}^{-i\mathbf{K}\cdot\mathbf{R}} \tag{4}$$

The discrete Fourier transforms on a generic one-index quantity $f$ are then

$$f(\mathbf{k}) = \sum_{\mathbf{r}} U_{\mathbf{k},\mathbf{r}}^\gamma f_{\mathbf{r}} \qquad f(\tilde{\mathbf{k}}) = \sum_{\tilde{\mathbf{r}}} U_{\tilde{\mathbf{k}},\tilde{\mathbf{r}}}^\Gamma f_{\tilde{\mathbf{r}}} \qquad f_{\mathbf{K}} = \sum_{\mathbf{R}} U_{\mathbf{K},\mathbf{R}}^c f_{\mathbf{R}} \tag{5}$$

or, in reverse,

$$f_{\mathbf{r}} = \sum_{\mathbf{k}} U_{\mathbf{k},\mathbf{r}}^{\gamma*} f(\mathbf{k}) \qquad f_{\tilde{\mathbf{r}}} = \sum_{\tilde{\mathbf{k}}} U_{\tilde{\mathbf{k}},\tilde{\mathbf{r}}}^{\Gamma*} f(\tilde{\mathbf{k}}) \qquad f_{\mathbf{R}} = \sum_{\mathbf{K}} U_{\mathbf{K},\mathbf{R}}^{c*} f_{\mathbf{K}} . \tag{6}$$

Quasi continuous indices, like $\mathbf{k}$ and $\tilde{\mathbf{k}}$, are most of the time indicated between parentheses.

These discrete Fourier transforms close by virtue of the following identities

$$\frac{1}{N} \sum_{\mathbf{k}} \mathrm{e}^{i\mathbf{k}\cdot\mathbf{r}} = \delta_{\mathbf{r}} \qquad\qquad \frac{1}{N} \sum_{\mathbf{r}} \mathrm{e}^{-i\mathbf{k}\cdot\mathbf{r}} = \Delta_\gamma(\mathbf{k}) \tag{7}$$

$$\frac{L}{N} \sum_{\tilde{\mathbf{k}}} \mathrm{e}^{i\tilde{\mathbf{k}}\cdot\tilde{\mathbf{r}}} = \delta_{\tilde{\mathbf{r}}} \qquad\qquad \frac{L}{N} \sum_{\tilde{\mathbf{r}}} \mathrm{e}^{-i\tilde{\mathbf{k}}\cdot\tilde{\mathbf{r}}} = \Delta_\Gamma(\tilde{\mathbf{k}}) \tag{8}$$

$$\frac{1}{L} \sum_{\mathbf{K}} \mathrm{e}^{i\mathbf{K}\cdot\mathbf{R}} = \delta_{\mathbf{R}} \qquad\qquad \frac{1}{L} \sum_{\mathbf{R}} \mathrm{e}^{-i\mathbf{K}\cdot\mathbf{R}} = \Delta_\gamma(\mathbf{K}) \tag{9}$$

where $\delta_{\mathbf{r}}$ is the usual Kronecker delta, used for all labels (since they are all discrete):

$$\delta_\alpha = \begin{cases} 1 & \text{if } \alpha = 0 \\ 0 & \text{otherwise} \end{cases} \qquad \delta_{\alpha\beta} \equiv \delta_{\alpha-\beta} \ , \tag{10}$$

and the $\Delta$'s are the so-called Laue functions:

$$\Delta_\gamma(\mathbf{k}) = \sum_{\mathbf{Q} \in \gamma^*} \delta_{\mathbf{k}+\mathbf{Q}} \tag{11}$$

$$\Delta_\Gamma(\tilde{\mathbf{k}}) = \sum_{\mathbf{P} \in \Gamma^*} \delta_{\tilde{\mathbf{k}}+\mathbf{P}} \ . \tag{12}$$

Laue functions are used instead of Kronecker deltas in momentum space because of the possibility of Umklapp processes. Note especially that even though

$$\delta_{\mathbf{k}} = \delta_{\tilde{\mathbf{k}}}\delta_{\mathbf{K}} \qquad (\mathbf{k} = \tilde{\mathbf{k}} + \mathbf{K}), \tag{13}$$

the same does not hold for the Laue functions:

$$\Delta_\gamma(\mathbf{k}) \neq \Delta_\Gamma(\tilde{\mathbf{k}})\Delta_\gamma(\mathbf{K}) \ . \tag{14}$$

Instead we have the following relations:

$$\Delta_\Gamma(\tilde{\mathbf{k}}) = \sum_{\mathbf{K}} \Delta_\gamma(\tilde{\mathbf{k}} + \mathbf{K}) \tag{15}$$

$$\Delta_\gamma(\mathbf{k}) = \Delta_\gamma(\tilde{\mathbf{k}} + \mathbf{K}) = \delta_{\tilde{\mathbf{k}}}\Delta_\gamma(\mathbf{K}) \tag{16}$$

which reflect the arbitrariness in the choice of Brillouin zone of the super-lattice (we use the term *Brillouin zone* in a rather liberal manner, as a complete and irreducible set of wave-vectors, and not as the Wigner-Seitz cell of the reciprocal lattice.)

A one-index quantity like the destruction operator $c_{\mathbf{r}} = c_{\tilde{\mathbf{r}}+\mathbf{R}}$ can be represented in a variety of ways, through partial Fourier transforms:

$$c_{\mathbf{R}}(\tilde{\mathbf{k}}) = \sum_{\tilde{\mathbf{r}}} U_{\mathbf{k}\tilde{\mathbf{r}}}^\Gamma \, c_{\tilde{\mathbf{r}}+\mathbf{R}} \tag{17}$$

$$c_{\tilde{\mathbf{r}},\mathbf{K}} = \sum_{\mathbf{R}} U_{\mathbf{K}\mathbf{R}}^c \, c_{\tilde{\mathbf{r}}+\mathbf{R}} \tag{18}$$

$$c_{\mathbf{K}}(\tilde{\mathbf{k}}) = \sum_{\tilde{\mathbf{r}},\mathbf{R}} U_{\mathbf{k}\tilde{\mathbf{r}}}^\Gamma U_{\mathbf{K}\mathbf{R}}^c \, c_{\tilde{\mathbf{r}}+\mathbf{R}} \tag{19}$$

$$c(\mathbf{k}) = \sum_{\mathbf{r}} U_{\mathbf{k}\mathbf{r}}^\gamma \, c_{\mathbf{r}} \tag{20}$$

The last two representations are not identical, since the phases in the two cases differ by $\tilde{\mathbf{k}} \cdot \mathbf{R}$. In other words, they are obtained respectively by applying the unitary matrices $\mathbf{S} \equiv \mathbf{U}^\Gamma \otimes \mathbf{U}^c$ and $\mathbf{U}^\gamma$ on the $\mathbf{r}$ basis, and these two operations are different, as the matrices $\mathbf{\Lambda} \equiv \mathbf{U}^\gamma \mathbf{S}^{-1}$ and $\mathbf{D} \equiv \mathbf{S}^{-1}\mathbf{U}^\gamma$ are not trivial:

$$\Lambda_{\mathbf{k}\mathbf{k}'} = \delta_{\tilde{\mathbf{k}}\tilde{\mathbf{k}}'} \frac{1}{L} \sum_{\mathbf{R}} e^{-i\mathbf{R}\cdot(\tilde{\mathbf{k}}+\mathbf{K}-\mathbf{K}')} \tag{21}$$

$$D_{\mathbf{r}\mathbf{r}'} = \delta_{\mathbf{R}\mathbf{R}'} \frac{L}{N} \sum_{\tilde{\mathbf{k}}} e^{i\tilde{\mathbf{k}}\cdot(\tilde{\mathbf{r}}-\tilde{\mathbf{r}}'-\mathbf{R})} \tag{22}$$

and one could write

$$c(\tilde{\mathbf{k}} + \mathbf{K}) = \sum_{\mathbf{K}'} \Lambda_{\mathbf{K}\mathbf{K}'}(\tilde{\mathbf{k}}) c_{\mathbf{K}'}(\tilde{\mathbf{k}}) \tag{23}$$

A two-index quantity like the hopping matrix $t_{\mathbf{rr}'}$ may thus have a number of different representations. Due to translation invariance on the lattice, this matrix is diagonal when expressed in momentum space: $t(\mathbf{k}, \mathbf{k}') = \varepsilon(\mathbf{k})\delta_{\mathbf{k},\mathbf{k}'}$, $\varepsilon(\mathbf{k})$ being the dispersion relation:

$$t_{\mathbf{rr}'} = \frac{1}{N} \sum_{\mathbf{k}} e^{i\mathbf{k}\cdot(\mathbf{r}-\mathbf{r}')} \varepsilon(\mathbf{k}) \tag{24}$$

However, we will very often use the mixed representation

$$t_{\mathbf{RR}'}(\tilde{\mathbf{k}}) = \sum_{\tilde{\mathbf{r}}} e^{i\tilde{\mathbf{k}}\cdot\tilde{\mathbf{r}}} t_{\mathbf{rr}'} \qquad \begin{cases} \mathbf{r} = \mathbf{R} \\ \mathbf{r}' = \tilde{\mathbf{r}} + \mathbf{R}' \end{cases} \tag{25}$$

For instance, if we tile the one-dimensional lattice with clusters of length $L = 2$, the nearest-neighbor hopping matrix, corresponding to the dispersion relation $\varepsilon(k) = -2t\cos(k)$, has the following mixed representation:

$$\mathbf{t}(\tilde{k}) = -t \begin{pmatrix} 0 & 1 + e^{-2i\tilde{k}} \\ 1 + e^{2i\tilde{k}} & 0 \end{pmatrix} \tag{26}$$

Finally, let us point out that the space $E$ of one-electron states is larger than the space of lattice sites $\gamma$, as it includes also spin and band degrees of freedom, which forms a set $B$ whose elements are indexed by $\sigma$. We could therefore write $E = \gamma \otimes B$. The transformation matrices defined above ($\mathbf{U}^{\gamma}$, $\mathbf{U}^{\Gamma}$ and $\mathbf{U}^{c}$) should, as necessary, be understood as tensor products ($\mathbf{U}^{\gamma} \otimes \mathbf{1}$, $\mathbf{U}^{\Gamma} \otimes \mathbf{1}$ and $\mathbf{U}^{c} \otimes \mathbf{1}$) acting trivially in $B$. This should be clear from the context.

# 3 Cluster perturbation theory

The simplest quantum cluster method is Cluster Perturbation Theory (CPT) [5,6]. CPT can be viewed as a cluster extension of strong-coupling perturbation theory [11], although limited to lowest order [12]. Its kinematic features are found in more sophisticated approaches like VCA or CDMFT, covered in sections 5 and 6.

## 3.1 Green functions

**The one-particle Green function** Quantum cluster methods are approximation strategies based on the one-particle Green function. Let us review basic concepts about this object. At zero-temperature, the Green function $G_{\mu\nu}(z)$ is a function of complex frequency $z$ defined as

$$G_{\mu\nu}(z) = \langle\Omega|c_{\mu}\frac{1}{z - H + E_0}c_{\nu}^{\dagger}|\Omega\rangle + \langle\Omega|c_{\nu}^{\dagger}\frac{1}{z + H - E_0}c_{\mu}|\Omega\rangle \quad, \tag{27}$$

where $|\Omega\rangle$ is the ground state (with energy $E_0$) associated with the Hamiltonian $H$, which includes the chemical potential. The indices $\mu, \nu$ stand for one-particle states, for instance a compound of site, spin and possibly orbital indices. $G_{\mu\nu}(z)$ contains dynamical information about one-particle excitations, such as the spectral weight measured in ARPES. We will generally use a boldface matrix notation ($\mathbf{G}$) for quantities carrying two one-body indices

$(G_{\mu\nu})$. A finite-temperature expression for the Green function (27) is obtained simply by replacing the ground state expectation value by a thermal average. Practical computations at finite temperature are mostly done using Monte Carlo methods, which rely on the path integral formalism and are performed as a function of imaginary time, not directly as a function of real frequencies. Since `pyqcm` is based on exact diagonalizations, we will confine ourselves to the zero-temperature formalism.

**Green function in the time domain**  The expression (27) may be unfamiliar to those used to a definition of the Green function in the time domain. Let us just mention the connection. We define the spectral function in the time domain and its Fourier transform as

$$A_{\mu\nu}(t) = \langle\{c_\mu(t), c_\nu^\dagger(0)\}\rangle \qquad A_{\mu\nu}(\omega) = \int_{-\infty}^\infty dt\, e^{i\omega t} A_{\mu\nu}(t) \tag{28}$$

where $\{\cdot, \cdot\}$ is the anticommutator and $z$ is a complex frequency. The time dependence is defined in the Heisenberg picture, i.e., $c_\mu(t) = e^{iHt}c_\mu(0)e^{-iHt}$. It can be shown that the Green function is related to $A_{\mu\nu}(z)$ by

$$G_{\mu\nu}(z) = \int_{-\infty}^\infty \frac{d\omega}{2\pi} \frac{A_{\mu\nu}(\omega)}{z - \omega} \quad . \tag{29}$$

The retarded Green function $G_{\mu\nu}^R(t)$ is defined, in the time domain, as

$$G_{\mu\nu}^R(t) = -i\Theta(t)\langle\{c_\mu(t), c_\nu^\dagger(0)\}\rangle = -i\Theta(t)A_{\mu\nu}(t) \tag{30}$$

where $\Theta(t)$ is the Heaviside step function. Since the Fourier transform of the latter is

$$\mathcal{F}(\Theta)(\omega) = \int_0^\infty dt\, e^{i\omega t} = i\frac{1}{\omega + i0^+} \quad , \tag{31}$$

a simple convolution shows that

$$G_{\mu\nu}^R(\omega) = \int_{-\infty}^\infty \frac{d\omega'}{2\pi} \frac{A_{\mu\nu}(\omega')}{\omega - \omega' + i0^+} = G_{\mu\nu}(\omega + i0^+) \quad . \tag{32}$$

In fact, this connection can be established easily from the spectral representation, introduced next.

**Spectral representation**  Let $\{|r\rangle\}$ be a complete set of eigenstates of $H$ with one particle *more* than the ground state, where $r$ is positive integer label. Likewise, let us use negative integer labels to denote eigenstates of $H$ with one particle *less* than the ground state. Then, by inserting completeness relations,

$$G_{\mu\nu}(z) = \sum_{r>0} \langle\Omega|c_\mu|r\rangle \frac{1}{z - E_r + E_0} \langle r|c_\nu^\dagger|\Omega\rangle + \sum_{r<0} \langle\Omega|c_\nu^\dagger|r\rangle \frac{1}{z + E_r - E_0} \langle r|c_\mu|\Omega\rangle \quad . \tag{33}$$

By setting

$$Q_{\mu r} = \begin{cases} \langle\Omega|c_\mu|r\rangle & (r > 0) \\ \langle r|c_\mu|\Omega\rangle & (r < 0) \end{cases} \qquad \text{and} \qquad \omega_r = \begin{cases} E_r - E_0 & (r > 0) \\ E_0 - E_r & (r < 0) \end{cases} \tag{34}$$

we write

$$G_{\mu\nu}(z) = \sum_r \frac{Q_{\mu r} Q_{\nu r}^*}{z - \omega_r} \quad . \tag{35}$$

This shows how the Green function is a sum over poles located at $\omega_r \in \mathbb{R}$, with residues that are products of overlaps of the ground state with energy eigenstates with one more ($\omega_r > 0$) or one less ($\omega_r < 0$) particle. The sum of residues is normalized to the unit matrix, as can be seen from the anticommutation relations:

$$
\sum_r Q_{\mu r} Q^*_{\nu r} = \sum_{r>0} \langle \Omega | c_\mu | r \rangle \langle r | c^\dagger_\nu | \Omega \rangle + \sum_{r<0} \langle \Omega | c^\dagger_\nu | r \rangle \langle r | c_\mu | \Omega \rangle
$$
$$
= \langle \Omega | \left( c_\mu c^\dagger_\nu + c^\dagger_\nu c_\mu \right) | \Omega \rangle = \delta_{\mu\nu} \quad .
$$
(36)

Thus, in the high-frequency limit, $\mathbf{G}(z \to \infty) = \mathbf{1}/z$ ($\mathbf{1}$ stands for the unit matrix).

The same procedure applied to the spectral function (28) leads to

$$
A_{\mu\nu}(\omega) = 2\pi \sum_r Q_{\mu r} Q^*_{\nu r} \, \delta(\omega - \omega_r)
$$
(37)

and this demonstrates the connection (29) between $A_{\mu\nu}(\omega)$ and $G_{\mu\nu}(z)$. The property (36) amounts to saying that $A_{\mu\mu}(\omega)$ is a probability density:

$$
A_{\mu\mu}(\omega) = 2\pi \sum_r |Q_{\mu r}|^2 \, \delta(\omega - \omega_r) \qquad \int_{-\infty}^{\infty} \frac{d\omega}{2\pi} A_{\mu\mu}(\omega) = 1
$$
(38)

The identity

$$
-\frac{1}{\pi} \operatorname{Im} \frac{1}{\omega + i0^+} = \delta(\omega)
$$
(39)

implies that

$$
A_{\mu\mu}(\omega) = -2 \operatorname{Im} G_{\mu\mu}(\omega + i0^+) \quad .
$$
(40)

From the definition of $Q_{\mu r}$, one sees that $A_{\mu\mu}(\omega)$ is the probability density for an electron added or removed from the ground state in the one-particle state $\mu$ to have an energy $\omega$. The density of states $\rho(\omega)$ is simply the trace

$$
\rho(\omega) = \frac{1}{N} \sum_\mu A_{\mu\mu}(\omega) = -\frac{2}{N} \operatorname{Im} \operatorname{tr} \mathbf{G}(\omega + i0^+)
$$
(41)

**Self-energy**   In the absence of interactions ($H_1 = 0$) the Hamiltonian reduces to

$$
H_0 = \sum_{\mu,\nu} t_{\mu\nu} c^\dagger_\mu c_\nu \quad .
$$
(42)

Since the matrix $\mathbf{t}$ is Hermitian, there exists a basis $\{|\ell\rangle\}$ of one-body states that makes it diagonal: $H_0 = \sum_\ell \epsilon_\ell c^\dagger_\ell c_\ell$. The ground state is then the filled Fermi sea:

$$
|\Omega\rangle = \prod_{\varepsilon_\ell < 0} c^\dagger_\ell |0\rangle
$$
(43)

and one-particle excited states are $c^\dagger_\ell |\Omega\rangle$ ($\varepsilon_\ell > 0$) with $E_\ell - E_0 = \varepsilon_\ell$ and $c_\ell |\Omega\rangle$ ($\varepsilon_\ell < 0$) with $E_\ell - E_0 = -\varepsilon_\ell$. The spectral representation is in that case extremely simple and the matrix $\mathbf{G} = \mathbf{G}_0$ is diagonal:

$$
G_{0,\ell\ell'}(z) = \frac{\delta_{\ell\ell'}}{z - \varepsilon_\ell}
$$
(44)

In any other basis of one-body states, in which $\mathbf{t}$ is not diagonal, the expression is simply

$$
\mathbf{G}_0(z) = \frac{1}{z - \mathbf{t}}
$$
(45)

In the presence of interactions, the Green function takes the following general form:

$$\mathbf{G}(z) = \frac{1}{z - \mathbf{t} - \mathbf{\Sigma}(z)} \tag{46}$$

where all the information related to $H_1$ is buried within the self-energy $\mathbf{\Sigma}(z)$. The relation (46), called *Dyson's equation*, may be regarded as a definition of the self-energy. It can be shown that the self-energy has a spectral representation similar to that of the Green function:

$$\Sigma_{\mu\nu}(z) = \Sigma_{\mu\nu}^{\infty} + \sum_r \frac{S_{\mu r} S_{\nu r}^*}{z - \sigma_r} \quad , \tag{47}$$

where the $\sigma_r$ are poles located on the real axis (they are zeros of the Green function). By contrast with the Green function, the self-energy may have a frequency-independent piece $\Sigma_{\mu\nu}^{\infty}$, which has the same effect as a hopping term; in fact, within the Hartree-Fock approximation, this is the only piece of the self-energy that survives.

**Averages of one-body operators**  Many physical observables are one-body operators, of the form

$$\hat{S} = \sum_{\mu,\nu} s_{\mu\nu} c_\mu^\dagger c_\nu \tag{48}$$

The ground state expectation value of such operators can be computed from the Green function $G_{\mu\nu}(z)$. Let us explain how.

From the spectral representation (35) of the Green function, we see that $\langle c_\mu^\dagger c_\nu \rangle$ is given by the integral of the Green function along a contour $C_<$ surrounding the negative real axis counterclockwise:

$$\langle c_\mu^\dagger c_\nu \rangle = \int_{C_<} \frac{\mathrm{d}z}{2\pi i} G_{\nu\mu}(z) \tag{49}$$

Therefore the expectation value we are looking for is

$$\bar{s} = \frac{1}{N} \langle \hat{S} \rangle = \frac{1}{N} \sum_{\mu,\nu} s_{\mu\nu} \langle c_\mu^\dagger c_\nu \rangle = \frac{1}{N} \int_{C_<} \frac{\mathrm{d}z}{2\pi i} \, \mathrm{tr} \, [\mathbf{s}\mathbf{G}(z)] \tag{50}$$

(we divide by $N$ to find an intensive quantity). The trace includes a sum over lattice sites, spin and band indices.

The contour $C_<$ can be taken as the imaginary axis (from $-iR$ to $iR$), plus the left semi-circle of radius $R$. Since $\mathbf{G}(z) \to \mathbf{1}/z$ as $z \to \infty$, the semi-circular part will contribute, but this contribution may be canceled by subtracting from $\mathbf{G}(z)$ a term like $\mathbf{1}/(z-p)$, with $p > 0$: the added term does not contribute to the integral, since its only pole lies outside the contour, yet it cancels the dominant $z^{-1}$ behavior as $z \to \infty$, leaving a contribution that vanishes on the semi-circle as $R \to \infty$. We are left with

$$\bar{s} = \frac{1}{N} \int_{-\infty}^{\infty} \frac{\mathrm{d}\omega}{2\pi} \left\{ \mathrm{tr} \, [\mathbf{s}\mathbf{G}(i\omega)] - \frac{\mathrm{tr} \, \mathbf{s}}{i\omega - p} \right\} \quad . \tag{51}$$

If the operator $\hat{S}$ is Hermitian, then so is the matrix $\mathbf{s}$. By virtue of the property $\mathbf{G}(z)^\dagger = \mathbf{G}(z^*)$, easily seen from (35), we have $\mathrm{tr} \, [\mathbf{s}\mathbf{G}(-i\omega)] = \mathrm{tr} \, [\mathbf{s}\mathbf{G}(i\omega)]^*$; this implies that $\bar{s}$ is real. Note that $\mathbf{s}$ can be expressed as a function of reduced wave-vector $\tilde{\mathbf{k}}$ and cluster indices (it is diagonal in $\tilde{\mathbf{k}}$ for a translation-invariant operator). The matrix $\mathbf{s}(\tilde{\mathbf{k}})$ is then $2L \times 2L$ (for a one-band model) and the above reduces to

$$\bar{s} = \frac{1}{N} \int_{-\infty}^{\infty} \frac{\mathrm{d}\omega}{2\pi} \sum_{\tilde{\mathbf{k}}} \left\{ \mathrm{tr} \left[ \mathbf{s}(\tilde{\mathbf{k}}) \mathbf{G}(i\omega) \right] - \frac{\mathrm{tr} \, \mathbf{s}(\tilde{\mathbf{k}})}{i\omega - p} \right\} \quad . \tag{52}$$

where the matrices involved are $2L \times 2L$.

## 3.2   Cluster Perturbation Theory

Cluster Perturbation Theory (CPT) proceeds as follows. First a cluster tiling is chosen (see, e.g., Fig. 1). Then the lattice Hamiltonian $H$ is written as $H = H_c + V$, where $H_c$ is the cluster Hamiltonian, obtained by severing the hopping terms between different clusters, whereas $V$ contains precisely those terms. $V$ is treated as a perturbation. It can be shown, by the techniques of strong-coupling perturbation theory [6, 12], that the lowest-order result for the Green function is

$$\mathbf{G}^{-1}(\omega) = \mathbf{G}_c^{-1}(\omega) - \mathbf{V} \ , \tag{53}$$

where $\mathbf{V}$ is the matrix of inter-cluster hopping terms and $\mathbf{G}_c(\omega)$ the exact Green function of the cluster only. This formula deserves a more thorough description: $\mathbf{G}$, $\mathbf{G}_c$ and $\mathbf{V}$ are matrices in the space $E$ one-electron states. This space is the tensor product $\gamma \otimes B$ of the lattice $\gamma$ by the space $B$ of band and spin states. For the remainder of this section we will ignore $B$, i.e., band and spin indices. In terms of compound cluster/cluster-site indices $(\tilde{\mathbf{r}}, \mathbf{R})$, $\mathbf{G}_c$ is diagonal in $\tilde{\mathbf{r}}$ and identical for all clusters, whereas $\mathbf{V}$ is essentially off-diagonal in $\tilde{\mathbf{r}}$. Because of translation invariance on the super-lattice, the above formula is simpler in terms of reduced wave-vectors, following a partial Fourier transform $\tilde{\mathbf{r}} \to \tilde{\mathbf{k}}$:

$$\mathbf{G}^{-1}(\tilde{\mathbf{k}}, \omega) = \mathbf{G}_c^{-1}(\omega) - \mathbf{V}(\tilde{\mathbf{k}}) \ . \tag{54}$$

The matrices appearing in the above formula are now of order $L$ (the number of sites in the cluster), i.e., they are matrices in cluster sites $\mathbf{R}$ only. $\mathbf{G}_c$ is independent of $\tilde{\mathbf{k}}$, whereas $\mathbf{V}$ is frequency independent.

The basic CPT relation (54) may also be expressed in terms of the self-energy $\mathbf{\Sigma}_c$ of the cluster Hamiltonian as

$$\mathbf{G}^{-1}(\tilde{\mathbf{k}}, \omega) = \mathbf{G}_0^{-1}(\tilde{\mathbf{k}}, \omega) - \mathbf{\Sigma}_c(\omega) \ , \tag{55}$$

where $\mathbf{G}_0(\tilde{\mathbf{k}}, \omega)$ is the Green function associated with the non-interacting part of the lattice Hamiltonian. This follows simply from the relations

$$\mathbf{G}_c^{-1} = \omega - \mathbf{t}_c - \mathbf{\Sigma}_c \tag{56}$$

$$\mathbf{G}_0^{-1} = \omega - \mathbf{t}_c - \mathbf{V} \ , \tag{57}$$

where $\mathbf{t}_c$ is the restriction to the cluster of the hopping matrix (chemical potential included). It is in the form (55) that CPT was first proposed [5].

## 3.3   Periodization

A supplemental ingredient to CPT is the periodization prescription, that provides a fully $\mathbf{k}$-dependent Green function out of the mixed representation $G_{\mathbf{RR}'}(\tilde{\mathbf{k}}, \omega)$. The cluster decomposition breaks the original lattice translation symmetry of the model. The Green function (54) is therefore not fully translation invariant and is not diagonal when expressed in terms of wave-vectors: $\mathbf{G} \to G(\mathbf{k}, \mathbf{k}')$. However, due to the residual super-lattice translation invariance, $\mathbf{k}'$ and $\mathbf{k}$ must map to the same wave-vector of the super-lattice Brillouin zone (or reduced Brillouin zone) and differ by an element of the reciprocal super-lattice. The periodization procedure proposed in Ref. [6] applies to the Green function itself:

$$G_{\text{per.}}(\mathbf{k}, \omega) = \frac{1}{L} \sum_{\mathbf{R}, \mathbf{R}'} e^{-i\mathbf{k} \cdot (\mathbf{R} - \mathbf{R}')} G_{\mathbf{RR}'}(\tilde{\mathbf{k}}, \omega) \ . \tag{58}$$

Since the reduced zone the wave-vector $\tilde{\mathbf{k}}$ is picked from is immaterial, on may replace $\tilde{\mathbf{k}}$ by $\mathbf{k}$ in the above formula (i.e. replacing $\tilde{\mathbf{k}}$ by $\tilde{\mathbf{k}} + \mathbf{K}$ yields the same result). This periodization formula may be heuristically justified as follows. In the $(\mathbf{K}, \tilde{\mathbf{k}})$ basis, the matrix $\mathbf{G}$ has the following form:

$$G_{\mathbf{K}\mathbf{K}'}(\tilde{\mathbf{k}}, \omega) = \frac{1}{L} \sum_{\mathbf{R},\mathbf{R}'} e^{-i(\mathbf{K}\cdot\mathbf{R} - \mathbf{K}'\cdot\mathbf{R}')} G_{\mathbf{R}\mathbf{R}'}(\tilde{\mathbf{k}}, \omega) \ . \tag{59}$$

This form can be further converted to the full wave-vector basis ($\mathbf{k} = \mathbf{K} + \tilde{\mathbf{k}}$) by use of the unitary matrix $\mathbf{\Lambda}$ of Eq (23):

$$
\begin{aligned}
G(\tilde{\mathbf{k}} + \mathbf{K}, \tilde{\mathbf{k}} + \mathbf{K}') &= \left( \mathbf{\Lambda}(\tilde{\mathbf{k}}) \mathbf{G} \mathbf{\Lambda}^\dagger(\tilde{\mathbf{k}}) \right)_{\mathbf{K}\mathbf{K}'} \\
&= \frac{1}{L^2} \sum_{\mathbf{R},\mathbf{R}',\mathbf{K}_1,\mathbf{K}_1'} e^{-i(\tilde{\mathbf{k}}+\mathbf{K}-\mathbf{K}_1)\cdot\mathbf{R}} e^{i(\tilde{\mathbf{k}}+\mathbf{K}'-\mathbf{K}_1')\cdot\mathbf{R}'} G_{\mathbf{K}_1 \mathbf{K}_1'} \\
&= \frac{1}{L} \sum_{\mathbf{R},\mathbf{R}'} e^{-i(\tilde{\mathbf{k}}+\mathbf{K})\cdot\mathbf{R}} e^{i(\tilde{\mathbf{k}}+\mathbf{K}')\cdot\mathbf{R}'} G_{\mathbf{R}\mathbf{R}'}(\tilde{\mathbf{k}}, \omega) \ .
\end{aligned}
\tag{60}
$$

The periodization prescription (58), or G-scheme, amounts to picking the diagonal piece of the Green function ($\mathbf{k} = \mathbf{k}'$) and discarding the rest. This makes sense in as much as the density of states $N(\omega)$ is the trace of the imaginary part of the Green function:

$$N(\omega) = -\frac{2}{N} \operatorname{Im} \operatorname{tr} \mathbf{G}(\omega) = -\frac{2}{N} \operatorname{Im} \sum_{\mathbf{r}} G_{\mathbf{r}\mathbf{r}}(\omega) = -\frac{2}{N} \operatorname{Im} \sum_{\mathbf{k}} G(\mathbf{k}, \omega) \tag{61}$$

and the spectral function $A(\mathbf{k}, \omega)$, as a partial trace, involves only the diagonal part. Indeed, it is a simple matter to show from the anticommutation relations that the frequency integral of the Green function is the unit matrix:

$$-2 \operatorname{Im} \int \frac{d\omega}{2\pi} \, \mathbf{G}(\omega) = \mathbf{1} \ . \tag{62}$$

This being representation independent, it follows that the frequency integral of the imaginary part of the off-diagonal components of the Green function vanishes.

As Fig. 2 shows, periodizing the Green function (Eq. (58)) reproduces the expected feature of the spectral function of the one-dimensional Hubbard model. In particular, the Mott gap that opens at arbitrary small $U$ (as known from the exact solution)

Another possible prescription for periodization is to apply the above procedure to the self-energy $\mathbf{\Sigma}_c$ instead. This is appealing since $\mathbf{\Sigma}_c$ is an irreducible quantity, as opposed to $\mathbf{G}$. This amounts to throwing out the off-diagonal components of $\mathbf{\Sigma}_c$ before applying Dyson's equation to get $\mathbf{G}$, as opposed to discarding the off-diagonal part at the last step, once the matrix inversion towards $\mathbf{G}$ has taken place. However, this periodization scheme leaves spectral weight within the Mott gap for arbitrary large value of $U$, which is clearly unphysical.

Yet another possibility is the cumulant periodization (or M-scheme), in which the first lattice cumulant of the Green function is periodized [13]. In practice, this proceeds as follows: The matrix of one-body terms is split into diagonal and off-diagonal parts:

$$\mathbf{t}(\tilde{\mathbf{k}}) = \mathbf{t}^{\text{diag}}(\tilde{\mathbf{k}}) + \mathbf{t}^{\text{off}}(\tilde{\mathbf{k}}) \tag{63}$$

We then proceed exactly like in the G-scheme, but without the off-diagonal piece of $\mathbf{t}$. In other words, we periodize the quantity

$$\mathbf{G}^{\text{diag}}(\tilde{\mathbf{k}}, \omega) = \left( \omega - \mathbf{t}^{\text{diag}}(\tilde{\mathbf{k}}) - \mathbf{\Sigma}(\omega) \right)^{-1} \tag{64}$$

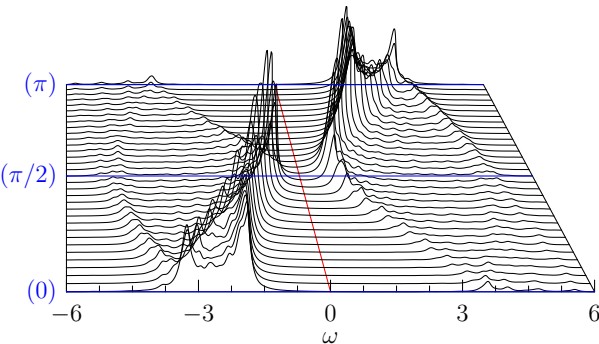

Figure 2: CPT spectral function of the one-dimensional, half-filled Hubbard model with $U = 4$, $t = 1$, with Green function periodization, from a 16-site cluster.

as in Eq. (58) and obtain $G_{\text{per.}}^{\text{diag}}(\mathbf{k}, \omega)$. We then express $\mathbf{t}^{\text{off}}(\tilde{\mathbf{k}})$ in the full Fourier representation ($t^{\text{off}}(\mathbf{k})$) and finally construct the periodized Green function

$$G_{\text{per.}}(\mathbf{k}, \omega) = \left[ G_{\text{per.}}^{\text{diag}}(\mathbf{k}, \omega)^{-1} - t^{\text{off}}(\mathbf{k}) \right]^{-1} \qquad \text{(M scheme)} \qquad (65)$$

### 3.4 General features of CPT

CPT has the following characteristics:

1. Although it is derived using strong-coupling perturbation theory, it is exact in the $U \to 0$ limit, as the self-energy disappears in that case.

2. It is also exact in the strong-coupling limit $t_{\mathbf{rr'}}/U \to 0$.

3. It provides an approximate lattice Green function for arbitrary wave-vectors. Hence its usefulness in comparing with ARPES data. Even though CPT does not have the self-consistency present in (C)-DMFT, at fixed computing resources it allows for the best momentum resolution. This is particularly important for the ARPES pseudogap in electron-doped cuprates that has quite a detailed momentum space structure, and for $d$-wave superconducting correlations where the zero temperature pair correlation length may extend beyond near-neighbor sites.

4. Although formulated as a lowest-order result of strong-coupling perturbation theory, it is not controlled by including higher-order terms in that perturbation expansion – this would be extremely difficult – but rather by increasing the cluster size.

5. It cannot describe broken-symmetry states. This is accomplished by VCA (Sect. 5) and CDMFT (Sect. 6).

## 4 Exact diagonalizations

Before going on to describe more sophisticated quantum cluster approaches, let us describe in some detail the method used by our library to compute the Green function of the cluster: The exact diagonalization method, based on the Lanczos algorithm and its variants. Note that the quantum cluster methods described here are not tied to a specific method for computing $\mathbf{G}_c$. For instance, Quantum Monte Carlo (QMC) or any other approximate method of solution for the cluster Green function could be used. The exact diagonalization

(ED) method has the advantage of being free from the fermion sign problem of QMC; moreover, the resulting Green function can be computed at arbitrary complex or real frequencies. On the other hand, it can only be applied to relatively small systems.

The basic idea behind exact diagonalization is one of brute force, but its practical implementation may require a lot of care depending on the desired level of optimization. Basically, an exact representation of the Hamiltonian action on arbitrary state vectors must be coded – this may or may not involve an explicit construction of the Hamiltonian matrix. Then the ground state is found in an quasi-exact way by an iterative method such as the Lanczos algorithm. The Green function is thereafter calculated by similar means to be described below. The main difficulty with execution is the large memory needed by the method, which grows exponentially with the number of degrees of freedom. As for coding, the main difficulty is to optimize the method, in particular by taking point group symmetries into account.

In this section, $H$ stands for the cluster (or impurity) Hamiltonian, and $\mathbf{G}(\omega)$ for the associated Green function, i.e., we omit the label $c$ used to distinguish cluster quantities from lattice ones.

## 4.1   Coding of the basis states

The first step in the exact diagonalization procedure is to define a coding scheme for the quantum basis states. A basis state may be specified by the occupation number $n_{i\sigma}$ ($= 0$ or 1) of electrons in the orbital labeled $i$ ($i = 1, \ldots, L$) of spin $\sigma = (\uparrow, \downarrow)$ and has the following expression in terms of creation operators:

$$(c_{1\uparrow}^{\dagger})^{n_{1\uparrow}} \cdots (c_{L\uparrow}^{\dagger})^{n_{L\uparrow}} (c_{1\downarrow}^{\dagger})^{n_{1\downarrow}} \cdots (c_{L\downarrow}^{\dagger})^{n_{L\downarrow}} |0\rangle \tag{66}$$

where the order in which the creation operators are applied is a matter of convention, but important. If the number of orbitals is smaller than or equal to 64, the string of occupation numbers $n_{i\sigma}$ forms the binary representation of a 64-bit unsigned integer $b$, which can be split into spin up and spin down parts:

$$b = b_{\uparrow} + 2^{L} b_{\downarrow} \tag{67}$$

There are $2^{2L}$ such states, but not all are relevant, since the Hubbard Hamiltonian is generally block-diagonal : The number of electrons of a given spin ($N_{\uparrow}$ and $N_{\downarrow}$) is often conserved and then commutes with the Hamiltonian $H$. Let us assume this situation holds for the moment. Then the exact diagonalization is to be performed in a sector (i.e. a subspace) of the total Hilbert space with fixed values of $N_{\uparrow}$ and $N_{\downarrow}$. This space has the tensor product structure

$$V = V_{N_{\uparrow}} \otimes V_{N_{\downarrow}} \tag{68}$$

and has dimension $d = d(N_{\uparrow})d(N_{\downarrow})$, where

$$d(N_{\sigma}) = \frac{L!}{N_{\sigma}!(L - N_{\sigma})!} \tag{69}$$

is the dimension of each factor, i.e., the number of ways to distribute $N_{\sigma}$ electrons among $L$ sites.

Note that the ground state $|\Omega\rangle$ of the Hamiltonian generally belongs to the sector $N_{\uparrow} = N_{\downarrow}$. For a half-filled, zero spin system ($N_{\uparrow} = N_{\downarrow} = L/2$), this translates into $d = (L!/(L/2)!^2)^2$, which behaves like $4^L/L$ for large $L$: The size of the eigen-problem grows exponentially with system size. By contrast, the non-interacting problem can be solved only by concentrating on one-electron states. For this reason, exact diagonalization

of the Hubbard Hamiltonian is restricted to systems of the order of 16 sites or less. Even though exact diagonalizations have been realized for the ground state of larger systems (e.g. $L = 22$), this operation can take weeks of computing time, whereas our goal here is to compute the Green function, not the ground state, and this is in repeated fashion as prescribed by embedding methods (VCA or CDMFT), in circumstances where particle number or spin is not conserved while exploring parameter space. Hence we cannot afford ED times that exceed a few minutes or hours.

In practice, a generic state vector is represented by an $d$-component array of double precision numbers. In order to apply or construct the Hamiltonian acting on such vectors, we need a way to translate the label of a basis state (an integer $i$ from 0 to $d-1$), into the binary representation (66). The way to do this depends on the level of complexity of the Hilbert space structure. In the simple case (68), one needs, for each spin, to build a two-way look-up table that tabulates the correspondence between consecutive integer labels and the binary representation of the spin up (resp. spin down) part of the basis state. Thus, given a binary representation $(b_\uparrow, b_\downarrow)$ of a basis state $|b\rangle = |b_\uparrow\rangle|b_\downarrow\rangle$, one immediately finds integer labels $I_\uparrow(b_\uparrow)$ and $I_\downarrow(b_\downarrow)$ and the label of the full basis state may be taken as

$$i = I_\uparrow(b_\uparrow) + d_{N\uparrow} I_\downarrow(b_\downarrow) \tag{70}$$

On the other hand, given a label $i$, the corresponding labels of each spin part are

$$i_\uparrow = \mod (i, d_{N\uparrow}) \qquad i_\downarrow = i/d_{N\uparrow} \tag{71}$$

where integer division (i.e. without fractional remainder) is used in the above expression. The binary representation $b$ is recovered by inverse tables $B$ as

$$b_\uparrow = B_\uparrow(i_\uparrow) \qquad b_\downarrow = B_\downarrow(i_\downarrow) \tag{72}$$

The next step is to construct the Hamiltonian matrix. The particular structure of the Hubbard model Hamiltonian brings a considerable simplification in the simple case studied here. Indeed, the Hamiltonian has the form

$$H = K_\uparrow \otimes 1 + 1 \otimes K_\downarrow + V_{\text{int.}} \tag{73}$$

where $K_\uparrow$ only acts on up electrons and $K_\downarrow$ on down electrons, and where the Coulomb repulsion term $V_{\text{int.}}$ is diagonal in the occupation number basis. Thus, storing the Hamiltonian in memory is not a problem : the diagonal $V_{\text{int.}}$ is stored (an array of size $d$), and the kinetic energy $K_\sigma$ (a matrix with $\sim Zd_\uparrow$ elements, $Z$ being the lattice coordination number) is stored in sparse form. Constructing this matrix, formally expressed as

$$K = \sum_{a,b} t_{ab} c_a^\dagger c_b \ , \tag{74}$$

needs some care with the signs. Basically, two basis states $|b_\sigma\rangle$ and $|b'_\sigma\rangle$ are connected with this matrix if their binary representations differ at two positions $a$ and $b$. The matrix element is then $(-1)^{M_{ab}} t_{ab}$, where $M_{ab}$ is the number of occupied sites between $a$ and $b$, i.e., assuming $a < b$,

$$M_{ab} = \sum_{c=a+1}^{b-1} n_c \tag{75}$$

For instance, the two states (10010110) and (10011100) with $L = 8$ are connected with the matrix element $-t_{46}$, where the sites are numbered from 0 to $L - 1$.

Computing the Hubbard interaction matrix elements is straightforward: a bit-wise AND is applied to the up and down parts of a binary state ($b_\uparrow \& b_\downarrow$ in C or C++) and the number of set bits of the result is the number of doubly occupied sites in that basis state.

If particle number and/or total spin projection is not conserved, then the decomposition (73) no longer applies. This occurs when studying superconductivity or including the spin-orbit coupling. We must deal with a much larger basis, and the correspondence between the index $I$ of a many-body state and its binary representation $b = B(I)$, while stored in memory, is less easily reversible; it is then more practical to binary-search the array $B$ for the value of the index $I$, given a binary state expression $b$.

## 4.2   The Lanczos algorithm for the ground state

Next, one must apply the exact diagonalization method *per se*, using the Lanczos algorithm. Generally, the Lanczos method [14] is used when one needs the extreme eigenvalues of a matrix too large to be fully diagonalized (e.g. with the Householder algorithm). The method is iterative and involves only the multiply-add operation from the matrix. This means in particular that the matrix does not necessarily have to be constructed explicitly, since only its action on a vector is needed. In some extreme cases where it is practical to do so, the matrix elements can be calculated 'on the fly', and this allows to save the memory associated with storing the matrix itself. On the other hand, storing the matrix in compressed sparse-row (CSR) format speeds up the multiply-add operation when it fits into memory. The optimal choice then depends on available resources and on the problem at hand.

The basic idea behind the Lanczos method is to build a projection $\mathscr{H}$ of the full Hamiltonian matrix $H$ onto the so-called Krylov subspace. Starting with a (random) state $|\phi_0\rangle$, the Krylov subspace is spanned by the iterated application of $H$:

$$\mathscr{K} = \text{span}\left\{|\phi_0\rangle, H|\phi_0\rangle, H^2|\phi_0\rangle, \cdots, H^M|\phi_0\rangle\right\} \tag{76}$$

the generating vectors above are not mutually orthogonal, but a sequence of mutually orthogonal vectors can be built from the following recursion relation

$$|\phi_{n+1}\rangle = H|\phi_n\rangle - a_n|\phi_n\rangle - b_n^2|\phi_{n-1}\rangle \tag{77}$$

where

$$a_n = \frac{\langle\phi_n|H|\phi_n\rangle}{\langle\phi_n|\phi_n\rangle} \qquad b_n^2 = \frac{\langle\phi_n|\phi_n\rangle}{\langle\phi_{n-1}|\phi_{n-1}\rangle} \qquad b_0 = 0 \tag{78}$$

and we set the initial conditions $b_0 = 0$, $|\phi_{-1}\rangle = 0$. At any given step, only three state vectors are kept in memory ($\phi_{n+1}$, $\phi_n$ and $\phi_{n-1}$). In the basis of normalized states $|n\rangle = |\phi_n\rangle/\sqrt{\langle\phi_n|\phi_n\rangle}$, the projected Hamiltonian has the tri-diagonal form

$$H^{(M)} = \begin{pmatrix} a_0 & b_1 & 0 & 0 & \cdots & 0 \\ b_1 & a_1 & b_2 & 0 & \cdots & 0 \\ 0 & b_2 & a_2 & b_3 & \cdots & 0 \\ \vdots & \vdots & \vdots & \vdots & \ddots & \vdots \\ 0 & 0 & 0 & 0 & \cdots & a_M \end{pmatrix} \tag{79}$$

Such a matrix is readily diagonalized by fast methods dedicated to tri-diagonal matrices and has eigen-pairs $(\lambda_i^{(M)}, \psi_i^{(M)})$ such that $H^{(M)}\psi_i^{(M)} = \lambda_i^{(M)}\psi_i^{(M)}$. If the eigenvalues $\lambda_i$ are sorted in ascending order, then a practical convergence criterion for the procedure is $b_M\psi_{M,M}^{(M-1)} < \delta$, where $\delta$ is a small tolerance, like $10^{-14}$ [14] (the largest and smallest

eigenvalues converge fastest in the Lanczos method). This may require a number $M$ of iterations between a few tens and $\sim 200$, depending on system size. In certain circumstances, for instance when the gap between the ground state and the first excited state is small, the number of required iterations may increase to several hundreds.

The ground state energy $E_0$ and the ground state $|\Omega\rangle$ are very well approximated by the lowest eigenvalue and the corresponding eigenvector of $H^{(M)}$. This provides us with the ground state $|\Omega\rangle$ in the reduced basis $\{|\phi_n\rangle\}$. But we need the ground state in the original basis, and this requires retracing the Lanczos iterations a second time – for the $|\phi_n\rangle$ are not stored in memory – and constructing the ground state progressively at each iteration from the known coefficients $\langle\Omega|\phi_n\rangle$.

The Lanczos procedure is simple and efficient. Convergence is fast if the lowest eigenvalue $E_0$ is well separated from the next one ($E_1$). If the ground state is degenerate ($E_1 = E_0$), the procedure will converge to a vector of the ground state subspace, a different one each time the initial state $|\phi_0\rangle$ is changed. Then another method, like the Davidson method [15], should be used; we will not describe it here, but it is available in the `pyqcm` library.

Note that the sequence of Lanczos vectors $|\phi_n\rangle$ is in principle orthogonal, as this is guaranteed by the three-way recursion relation (77). However, numerical errors will introduce 'orthogonality leaks', and after a few tens of iterations the Lanczos basis will become over-complete in the Krylov subspace. This will translate in multiple copies of the ground state eigenvalue in the tri-diagonal matrix (79), which should not be taken as a true degeneracy. However, as long as one is only interested in the ground state and not in the multiplicity of the lowest eigenvalues, this is not a problem.

## 4.3   The Lanczos algorithm for the Green function

Once the ground state is known, the cluster Green function $\mathbf{G}_c(z)$ may be computed. The zero-temperature Green function $G_{\mu\nu}(z)$ has the following expression, as a function of the complex-valued frequency $z$:

$$G_{\mu\nu}(z) = G_{\mu\nu}^{(e)}(z) + G_{\mu\nu}^{(h)}(z) \tag{80}$$

$$G_{\mu\nu}^{(e)}(z) = \langle\Omega|c_\mu \frac{1}{z - H + E_0} c_\nu^\dagger|\Omega\rangle \tag{81}$$

$$G_{\mu\nu}^{(h)}(z) = \langle\Omega|c_\nu^\dagger \frac{1}{z + H - E_0} c_\mu|\Omega\rangle \tag{82}$$

Here the indices $\mu, \nu$ are composite indices for site and spin. In the basic Hubbard model (1), spin is conserved and we need only to consider the creation and annihilation of up-spin electrons.

Let us first describe the simple Lanczos method for computing the Green function, which provides a continued-fraction representation of its frequency dependence. Consider first the function $G_{\mu\nu}^{(e)}(z)$. One needs to know the action of $(z - H + E_0)^{-1}$ on the state $|\phi_\mu\rangle = c_\mu^\dagger|\Omega\rangle$, and then calculate

$$G_{\mu\mu}^{(e)} = \langle\phi_\mu| \frac{1}{z - H + E_0} |\phi_\mu\rangle \tag{83}$$

As with any generic function of $H$, this one can be expanded in powers of $H$:

$$\frac{1}{z - H} = \frac{1}{z} + \frac{1}{z^2} H + \frac{1}{z^3} H^2 + \cdots \tag{84}$$

and the action of this operator can be evaluated exactly at order $H^M$ in a Krylov subspace (76). Thus we again resort to the Lanczos algorithm: A Lanczos sequence is calculated

from the initial, normalized state $|\phi_0\rangle = |\phi_\mu\rangle/b_0$ where $b_0^2 = \langle\phi_\mu|\phi_\mu\rangle$. This sequence generates a tri-diagonal representation of $H$, albeit in a different Hilbert space sector : that with $N_\uparrow+1$ up-spin electrons and $N_\downarrow$ down-spin electrons. Once the preset maximum number of Lanczos steps has been reached (or a small enough value of $b_n$), the tri-diagonal representation (79) may then be used to calculate (83). This amounts to the matrix element $b_0^2[(z - H + E_0)^{-1}]_{00}$ (the first element of the inverse of a tri-diagonal matrix), which has a simple continued fraction form [16]:

$$G_{\mu\mu}^{(e)}(z) = \cfrac{b_0^2}{z - a_0 - \cfrac{b_1^2}{z - a_1 - \cfrac{b_2^2}{z - a_2 - \cdots}}} \tag{85}$$

Thus, evaluating the Green function, once the arrays $\{a_n\}$ and $\{b_n\}$ have been found, reduces to the calculation of a truncated continued fraction, which can be done recursively in $M$ steps, starting from the bottom floor of the fraction.

Consider next the case $\mu \neq \nu$. The continued fraction representation applies only to the case where the same state $|\phi\rangle$ appears on the two sides of (83). If $\mu \neq \nu$, this is no longer the case, but we may use the following trick : we define the combination

$$G_{\mu\nu}^{(e)+}(z) = \langle\Omega|(c_\mu + c_\nu)\frac{1}{z - H + E_0}(c_\mu + c_\nu)^\dagger|\Omega\rangle \tag{86}$$

If the Hamiltonian is real, we can use the symmetry $G_{\mu\nu}^{(e)}(z) = G_{\nu\mu}^{(e)}(z)$ and this leads to

$$G_{\mu\nu}^{(e)}(z) = \frac{1}{2}(G_{\mu\nu}^{(e)+}(z) - G_{\mu\mu}^{(e)}(z) - G_{\nu\nu}^{(e)}(z)) \tag{87}$$

where $G_{\mu\nu}^{(e)+}$ can be calculated in the same way as $G_{\mu\mu}^{(e)}$, i.e., with a simple continued fraction. For a complex-valued Hamiltonian one additional step is needed: we also compute

$$G_{\mu\nu}^{(e)+i}(z) = \langle\Omega|(c_\mu + ic_\nu)\frac{1}{z - H + E_0}(c_\mu + ic_\nu)^\dagger|\Omega\rangle \tag{88}$$

It is then a simple matter to show that

$$G_{\mu\nu}^{(e)}(z) = \frac{1}{2}\left[G_{\mu\nu}^{(e)+}(z) + iG_{\mu\nu}^{(e)+i}(z) - (1 + i)(G_{\mu\mu}^{(e)}(z) + G_{\nu\nu}^{(e)}(z))\right] \tag{89}$$

and we must use the property that $G_{\nu\mu}^*(z^*) = G_{\mu\nu}(z)$, valid for $G^{(e,h)}$ separately. Note that the continued fraction coefficients are all real, even for a complex Hamiltonian, so that $G_{\mu\nu}^{(e)+}(z)^* = G_{\mu\nu}^{(e)+}(z^*)$ and $G_{\mu\nu}^{(e)+i}(z)^* = G_{\mu\nu}^{(e)+i}(z^*)$. We proceed likewise for $G_{\mu\nu}^{(h)+}(z)$ and $G_{\mu\nu}^{(h)+i}(z)$.

Thus, the cluster Green function is encoded in $L(L + 1)$ or $L^2$ continued fractions for the real and complex cases respectively. Their coefficients are stored in memory, so that $\mathbf{G}(z)$ can be computed on demand for any complex frequency $z$.

Note that a minimal way to take advantage of cluster symmetries is to restrict the calculation of the Green function to an irreducible set of pairs $(\mu, \nu)$ of orbitals that can generate all other pairs by symmetry operations of the cluster. Thus, if a symmetry operation $g$ takes the orbital $\mu$ into the orbital $g(\mu)$, we have

$$G_{\mu\nu}(z) = G_{g(\mu)g(\nu)}(z) \tag{90}$$

Taking this into account is an easy and important time saver, but not as efficient as using a basis of symmetry eigenstates, as described in Sect. 4.5 below.

## 4.4   The band Lanczos algorithm for the Green function

An alternate way of computing the cluster Green function is to apply the *band Lanczos* procedure [17]. This is a generalization of the Lanczos procedure in which the Krylov subspace is spanned not by one, but by many states. Let us assume that up and down spins are decoupled, so that the Green function is $L \times L$ block diagonal. The $L$ states $|\phi_\mu\rangle = c_\mu^\dagger |\Omega\rangle$ are first constructed, and then one builds the projection $\mathscr{H}$ of $H$ on the Krylov subspace spanned by

$$\left\{ |\phi_1\rangle, \ldots, |\phi_L\rangle, H|\phi_1\rangle, \ldots, H|\phi_L\rangle, \ldots, H^M|\phi_1\rangle, \ldots, H^M|\phi_L\rangle \right\} \tag{91}$$

A Lanczos basis $\{|n\rangle\}$ is constructed by successive application of $H$ and orthonormalization with respect to the previous $2L$ basis vectors. In principle, each new basis vector $|n\rangle$ is already automatically orthogonal to basis vectors $|1\rangle$ through $|n - 2L - 1\rangle$, although 'orthogonality leaks' arise eventually and may be problematic. A practical rule of thumb to avoid these problems is to control the number $M$ of iterations by the convergence of the lowest eigenvalue of $\mathscr{H}$ (e.g. to one part in $10^{10}$). Independently of this, one must be careful about potential redundant basis vectors in the Krylov subspace, which must be properly 'deflated' [17]. The number of states $R$ in the Krylov subspace at convergence is typically between 100 and 500, depending on system size. The $R \times R$ matrix $\mathscr{H}$, which is tri-diagonal in the ordinary Lanczos method, now is a banded matrix made of $2L$ diagonals around the central diagonal. It is then a simple matter to obtain a Lehmann representation of the Green function in the Krylov subspace by computing the projections $Q_{\mu r}$ of $|\phi_\mu\rangle$ on the eigenstates of $\mathscr{H}$ (the inner products of the $|\phi_\mu\rangle$'s with the Lanczos vectors are calculated at the same time as the latter are constructed). The Green function can then be expressed in a Lehmann representation (35). The two contributions $G_{\mu\nu}^{(e)}$ and $G_{\mu\nu}^{(h)}$ to the Green function are computed separately, and the corresponding matrices $\mathbf{Q}$ and $\boldsymbol{\Lambda}$ are simply concatenated to form the complete $\mathbf{Q}$- and $\boldsymbol{\Lambda}$-matrices, which are then stored and allow again for a quick calculation of the Green function as a function of the complex frequency $z$, following Eq. (35). The matrix $2L \times R$ matrix $\mathbf{Q}$ has the property that

$$\mathbf{Q}\mathbf{Q}^\dagger = \mathbf{1}_{2L \times 2L} \tag{92}$$

This holds even if the Lehmann representation is obtained from a subspace and not the full space, and is simply a consequence of the anticommutation relations $\{c_\mu, c_\nu^\dagger\} = \delta_{\mu\nu}$.

The band Lanczos method requires more memory than the usual Lanczos method, since $2L + 1$ vectors must simultaneously be kept in memory, compared to 3 for the simple Lanczos method. On the other hand, it is faster since all pairs $(\mu, \nu)$ are covered in a single procedure, compared to $L(L + 1)/2$ procedures in the simple Lanczos method. Thus, we gain a factor $L^2$ in speed at the cost of a factor $L$ in memory. Another advantage is that it provides a Lehmann representation of the Green function.

## 4.5   Cluster symmetries

It is possible to optimize the exact diagonalization procedure by taking advantage of the symmetries of the cluster Hamiltonian, in particular coming from cluster geometry. If the Hamiltonian is invariant under a discrete group $\mathfrak{G}$ of symmetry operations and $|\mathfrak{G}|$ denotes the number of such elements (the order of the group), the dimension of the largest Hilbert space needed can be reduced by a factor of almost $|\mathfrak{G}|$, and the number of state vectors needed in the band Lanczos method reduced by the same factor. The corresponding speed gain is appreciable. The price to pay is a higher complexity in coding the basis states, which almost forces one to store the Hamiltonian matrix in memory, if it were not already,

since computing matrix elements 'on the fly' becomes more time consuming. Note that we are using open boundary conditions, and therefore there is no translation symmetry within the cluster; thus we are concerned with points groups, not space groups.

Let us start with a simple example: a cluster invariant with respect to a single mirror symmetry, or a single rotation by $\pi$. One may think of a one-dimensional cluster, for instance, with a left-right mirror symmetry. The corresponding symmetry group is $C_2$, with two elements: the identity $e$ and the inversion $\iota$. The group $C_2$ contains two irreducible representations, noted $A$ and $B$, corresponding respectively to states that are even and odd with respect to $\iota$. Because the Hamiltonian is invariant under inversion: $H = \iota^{-1}H\iota$, eigenvectors of $H$ will be either even or odd, i.e. belong either to the A or to the B representation. Likewise, the Hamiltonian will have no matrix elements between states belonging to different representations.

In order to take advantage of this fact, one needs to construct a basis containing only states of a given representation. The occupation number basis states $|b\rangle$ (or binary states, as we will call them) introduced above are no longer adequate. In the case of the simple group $C_2$, one should rather consider the even and odd combinations $|b\rangle \pm \iota|b\rangle$ (and some of these combinations may vanish). Yet we still need a scheme to label the different basis states and have a quick access to their occupation number representation, which allows us to compute matrix elements. Let us briefly describe how this can be done (a more detailed discussion can be found, e.g., in Ref. [18]). Under the action of the group $\mathfrak{G}$, each binary state generates an 'orbit' of binary states, whose length is the order $|\mathfrak{G}|$ of the group, or a divisor thereof. To such an orbit correspond at most $d_\alpha$ states in the irreducible representation labeled $\alpha$, given by the corresponding projection operator:

$$|\psi\rangle = \frac{d_\alpha}{|\mathfrak{G}|} \sum_{g \in \mathfrak{G}} \chi_g^{(\alpha)*} g|b\rangle \tag{93}$$

where $d_\alpha$ is the dimension of the irreducible representation $\alpha$ and $\chi_g^{(\alpha)}$ the group character associated with the group element $g$ (or rather its conjugacy class) and representation $\alpha$.

We will restrict the discussion to the case of Abelian groups, of which all irreducible representations are one-dimensional ($d_\alpha = 1$; the case $d_\alpha > 1$ turns out to be quite a bit more complex). Then the state $|\psi\rangle$ is either zero or unique for a given orbit. We can then select a representative binary state for each orbit (e.g. the one associated with the smallest binary representation) and use it as a label for the state $|\psi\rangle$. We still need an index function $B(i)$ which provides the representative binary state for each consecutive label $i$. The reverse correspondence $i = I(b)$ is trickier, since symmetrized states are no longer factorized as products of up and down spin parts. It is better then to binary-search the array $B$ for the value of the index $i$ that provides a given binary state $b$.

Once the basis has been constructed, one needs to construct a matrix representation of the Hamiltonian in that representation. Given two states $|\psi_1\rangle$ and $|\psi_2\rangle$, represented by the binary states $|b_1\rangle$ and $|b_2\rangle$, it is a simple matter to show that the matrix element is

$$\langle \psi_2|H|\psi_1\rangle = \frac{d_\alpha}{|\mathfrak{G}|} \sum_{g \in \mathfrak{G}} \chi_g^{(\alpha)*} \phi_g(b) \langle gb_2|H|b_1\rangle \tag{94}$$

where the phase $\phi_g(b)$ is defined by the relation

$$g|b\rangle = \phi_g(b)|gb\rangle \ . \tag{95}$$

In the above relation, $|gb\rangle$ is the binary state obtained by applying the symmetry operation $g$ to the occupation numbers forming $b$, whereas the phase $\phi_g(b)$ is the product of

Table 1: Number of matrix elements of a given value in the nearest-neighbor hopping operator on the half-filled $3 \times 4 = 12$ site cluster, for each irreducible representation of $C_{2v}$. The dimension of each subspace is indicated on the second row.

|  | $A_1$ | $A_2$ | $B_1$ | $B_2$ |
|---|---|---|---|---|
| dim. | $213,840$ | $213,248$ | $213,440$ | $213,248$ |
| value |  |  |  |  |
| $-2$ | $96$ | $736$ | $704$ | $0$ |
| $-\sqrt{2}$ | $12,640$ | $6,208$ | $7,584$ | $5,072$ |
| $-1$ | $2,983,264$ | $2,936,144$ | $2,884,832$ | $2,911,920$ |
| $1$ | $952,000$ | $997,168$ | $1,050,432$ | $1,021,392$ |
| $\sqrt{2}$ | $5,088$ | $2,304$ | $3,232$ | $2,992$ |
| $2$ | $32$ | $0$ | $0$ | $0$ |

signs collected from all the permutations of creation operators needed to go from $b$ to $gb$. Formula (94) is used as follows to construct the Hamiltonian matrix: First, the Hamiltonian can be written as $H = \sum_r H_r$, where $H_r$ is a hopping term between specific sites, or a diagonal term like the interaction. One then loops over all $b_1$'s. For each $b_1$, and each term $H_r$, one constructs the single binary state $H_r|b_1\rangle$. One then finds the representative $b_2$ of that binary state, by applying on it all possible symmetry operations until $g$ is found such that $|gb_2\rangle = H_r|b_1\rangle$. During this operation, the phase $\phi_g(b)$ must also be collected. Then the matrix element (94) is added to the list of stored matrix elements. Since each term $H_r$ individually is not invariant under the group, there will be more matrix elements generated than there should be, i.e., there will be cancellations between different matrix elements associated with the same pair $(b_1, b_2)$ and produced by the different $H_r$'s. For this reason, it is useful to first store all matrix elements associated with a given $b_1$ in an intermediate location in order for the cancellations to take effect, and then to store the cleaned up 'column' labeled by $b_1$ to its definitive storage location. Needless to say, one should only store the row and column indices of each element of a given value.

Table 1 gives the values and number of matrix elements found for the nearest-neighbor hopping terms on the half-filled 12-site ($3 \times 4$) cluster, in each of the four irreducible representations of the group $C_{2v}$.

## 4.6   Green functions using cluster symmetries

Most of the time, the ground state lies in the trivial (symmetric) representation. However, taking advantage of symmetries in the calculation of the Green function requires all the irreducible representations to be included in the calculation. Consider for instance the simple example of a $C_2$ symmetry, with a ground state $|\Omega\rangle$ in the $A$ (even) representation. Constructing the Green function involves applying on $|\Omega\rangle$ the destruction operator $c_a$ (or the creation operator $c_a^\dagger$) associated to site $a$. The excited state thus produced does not belong to a well-defined representation. Instead, one should destroy (or create) and electron in an odd or even state, by using the linear combinations $c_a \pm c_{\iota a}$, where $\iota a$ is the site obtained by applying the symmetry operation to $a$. Thus, in computing the Green function (80), one should express each creation/destruction operator in terms of symmetrized combinations, e.g.,

$$c_a = \frac{1}{2}(c_a + c_{\iota a}) + \frac{1}{2}(c_a - c_{\iota a}) \tag{96}$$

More generally, one would use symmetrized combinations of operators

$$c_\rho^{(\alpha)} = \sum_a M_{\rho a}^{(\alpha)} c_a \tag{97}$$

such that $c_\rho^{(\alpha)}$ transforms under representation $\alpha$, and $\rho$ labels the different possibilities. For instance, for a linear cluster of length 4 and an inversion symmetry that maps the sites (1234) into (4321), these operators are

$$\begin{aligned} c_1^{(A)} &= c_1 + c_4 & c_1^{(B)} &= c_1 - c_4 \\ c_2^{(A)} &= c_2 + c_3 & c_2^{(B)} &= c_2 - c_3 \end{aligned} \tag{98}$$

Then, for each representation, one may use the band Lanczos procedure and obtain a Lehmann representation $Q_{\rho r}^{(\alpha)}$ for the associated Green function $G_{\rho\sigma}^{(\alpha)}(z)$. If the ground state is in representation $\alpha$ and the operators $c_\rho^{(\beta)}$ of representation $\beta$ are used, the Hilbert space sector to work with will be the tensor product representation $\alpha \otimes \beta$, which poses no problem at all when all irreducible representations are one-dimensional, but would bring additional complexity if the ground state were in a multidimensional representation. Finally, one may bring together the different pieces, by building a $L \times L$ matrix $M_{\rho a}$ that is the vertical concatenation of the various rectangular matrices $M_{\rho a}^{(\alpha)}$, and returning to the usual $\mathbf{Q}$-matrix representation

$$Q_{ar} = (\mathbf{M}^{-1})_{a\rho} Q_{\rho r} \tag{99}$$

Using cluster symmetries for the Green function saves a factor $|\mathfrak{G}|$ in memory because of the reduction of the Hilbert space dimension, and an additional factor of $|\mathfrak{G}|$ since the number of input vectors in the band Lanczos procedure is also divided by $|\mathfrak{G}|$. Typically then, most of the memory will be used to store the Hamiltonian matrix.

## 5 The Variational Cluster Approximation

### 5.1 The self-energy functional approach

That CPT is incapable of describing broken symmetries is its major drawback. Treating spontaneously broken symmetries requires some sort of self-consistent procedure, or a variational principle. Ordinary mean-field theory does precisely that, but is limited by its discarding of fluctuations and its uncontrolled character.

A heuristic way of treating broken symmetry states within CPT would be to add to the cluster Hamiltonian $H_c$ a Weiss field that pushes the system towards some predetermined form of order. For instance, the following term, added to the Hamiltonian, would induce Néel antiferromagnetism:

$$H_M = M\hat{M} \qquad , \qquad \hat{M} = \sum_{\mathbf{R}} e^{i\mathbf{Q}\cdot\mathbf{R}}(n_{\mathbf{R}\uparrow} - n_{\mathbf{R}\downarrow}) \tag{100}$$

where $\mathbf{Q} = (\pi, \pi)$ is the antiferromagnetic wave-vector. What is needed is a procedure to set the value of the Weiss parameter $M$. Adopting a mean-field-like procedure (i.e. factorizing the interaction in the correct channel and applying a self-consistency condition) would bring us exactly back to ordinary mean-field theory: the interaction having disappeared, the cluster decomposition would be suddenly useless and CPT would provide the same result regardless of cluster size.

Figure 3: Diagrammatic definition of the Luttinger-Ward functional, as a sum over two-particle irreducible graphs.

The solution to that conundrum is most elegantly provided by the self-energy functional approach (SFA), proposed by Potthoff [19, 20]. This approach also has the merit of presenting various cluster schemes from a unified point of view. It can also be seen as a special case of the more general inversion method [21], reviewed in Ref. [22] in the context of Density Functional Theory and DMFT.

To start with, let us introduce a functional $\Omega_{\mathbf{t}}[\mathbf{G}]$ of the Green function:

$$\Omega_{\mathbf{t}}[\mathbf{G}] = \Phi[\mathbf{G}] - \mathrm{Tr}((\mathbf{G}_{0\mathbf{t}}^{-1} - \mathbf{G}^{-1})\mathbf{G}) + \mathrm{Tr}\ln(-\mathbf{G}). \tag{101}$$

This means that, given any Green function $G_{ij}(\omega)$ one can cook up – yet with the usual analytic properties of Green functions as a function of frequency – this expression yields a number. In the above expression, products and powers of Green functions – e.g. in series expansions like that of the logarithm – are to be understood in a functional matrix sense. This means that position $i$ and time $\tau$, or equivalently, position and frequency, are merged into a single index. Accordingly, the symbol Tr denotes a functional trace, i.e., it involves not only a sum over sites indices, but also over frequencies. The latter can be taken as a sum over Matsubara frequencies at finite temperature, or as an integral over the imaginary frequency axis at zero temperature.

The Luttinger Ward functional $\Phi[\mathbf{G}]$ entering this expression is usually defined as the sum of *two-particle irreducible* (2PI) diagrams : diagrams that cannot by split into disjoint parts by cutting two fermion lines (Fig. 3). These are sometimes called *skeleton* diagrams, although 'two-particle irreducible' is more accurate. A diagram-free definition of $\Phi[\mathbf{G}]$ is also given in Ref. [23]. For our purposes, what is important is that (1) The functional derivative of $\Phi[\mathbf{G}]$ is the self-energy

$$\frac{\delta\Phi[\mathbf{G}]}{\delta\mathbf{G}} = \mathbf{\Sigma} \tag{102}$$

(as defined diagrammatically) and (2) it is a universal functional of $\mathbf{G}$ in the following sense: whatever the form of the one-body Hamiltonian, it depends only on the interaction and, functionally, it has the same dependence on $\mathbf{G}$. This is manifest from its diagrammatic definition, since only the interaction (dotted lines) and the Green function given as argument, enter the expression. The dependence of the functional $\Omega_{\mathbf{t}}[\mathbf{G}]$ on the one-body part of the Hamiltonian is denoted by the subscript $\mathbf{t}$ and it comes only through $\mathbf{G}_{0\mathbf{t}}^{-1} = \omega - \mathbf{t}$ appearing on the right-hand side of Eq. (101).

The functional $\Omega_{\mathbf{t}}[\mathbf{G}]$ has the important property that it is stationary when $\mathbf{G}$ takes the value prescribed by Dyson's equation. Indeed, given the last two equations, the Euler equation takes the form

$$\frac{\delta\Omega_{\mathbf{t}}[\mathbf{G}]}{\delta\mathbf{G}} = \mathbf{\Sigma} - \mathbf{G}_{0\mathbf{t}}^{-1} + \mathbf{G}^{-1} = 0. \tag{103}$$

This is a dynamic variational principle since it involves the frequency appearing in the Green function, in other words excited states are involved in the variation. At this stationary point, and only there, $\Omega_{\mathbf{t}}[\mathbf{G}]$ is equal to the physical (thermodynamic) grand potential.

Contrary to Ritz's variational principle, this last equation does not tell us whether $\Omega_{\mathbf{t}}[\mathbf{G}]$ is a minimum, a maximum, or a saddle point there.

There are various ways to use the variational principle described above. The most common one is to approximate $\Phi[\mathbf{G}]$ by a finite set of diagrams. This is how one obtains the Hartree-Fock, the FLEX approximation [24] or other so-called thermodynamically consistent theories. This is what Potthoff calls a type II approximation strategy. [25] A type I approximation simplifies the Euler equation itself. In a type III approximation, one uses the exact form of $\Phi[\mathbf{G}]$ but only on a limited domain of trial Green functions.

Following Potthoff, we adopt the type III approximation on a functional of the self-energy instead of on a functional of the Green function. Suppose we can locally invert Eq. (102) for the self-energy to write $\mathbf{G}$ as a functional of $\boldsymbol{\Sigma}$. We can use this result to write,

$$\Omega_{\mathbf{t}}[\boldsymbol{\Sigma}] = F[\boldsymbol{\Sigma}] - \mathrm{Tr}\ln(-\mathbf{G}_{0\mathbf{t}}^{-1} + \boldsymbol{\Sigma}). \tag{104}$$

where we defined

$$F[\boldsymbol{\Sigma}] = \Phi[\mathbf{G}] - \mathrm{Tr}(\boldsymbol{\Sigma}\mathbf{G}). \tag{105}$$

and where it is implicit that $\mathbf{G} = \mathbf{G}[\boldsymbol{\Sigma}]$ is now a functional of $\boldsymbol{\Sigma}$. $F[\boldsymbol{\Sigma}]$, along with the expression (102) for the derivative of the Luttinger-Ward functional, defines the Legendre transform of the Luttinger-Ward functional. It is easy to verify that

$$\frac{\delta F[\boldsymbol{\Sigma}]}{\delta\boldsymbol{\Sigma}} = \frac{\delta\Phi[\mathbf{G}]}{\delta\mathbf{G}}\frac{\delta\mathbf{G}[\boldsymbol{\Sigma}]}{\delta\boldsymbol{\Sigma}} - \boldsymbol{\Sigma}\frac{\delta\mathbf{G}[\boldsymbol{\Sigma}]}{\delta\boldsymbol{\Sigma}} - \mathbf{G} = -\mathbf{G} \tag{106}$$

hence, $\Omega_{\mathbf{t}}[\boldsymbol{\Sigma}]$ is stationary with respect to $\boldsymbol{\Sigma}$ when Dyson's equation is satisfied

$$\frac{\delta\Omega_{\mathbf{t}}[\boldsymbol{\Sigma}]}{\delta\boldsymbol{\Sigma}} = -\mathbf{G} + (\mathbf{G}_{0\mathbf{t}}^{-1} - \boldsymbol{\Sigma})^{-1} = 0. \tag{107}$$

To perform a type III approximation on $F[\boldsymbol{\Sigma}]$, we take advantage that it is universal, i.e., that it depends only on the interaction part of the Hamiltonian and not on the one-body part. We then consider another Hamiltonian, denoted $H'$ and called the *reference system*, that describes the same degrees of freedom as $H$ and shares the same interaction (i.e. two-body) part. Thus $H$ and $H'$ differ only by one-body terms. We have in mind for $H'$ the cluster Hamiltonian, or rather the sum of all (mutually decoupled) cluster Hamiltonians. At the physical self-energy $\boldsymbol{\Sigma}$ of the cluster, Eq. (104) allows us to write

$$\Omega_{\mathbf{t}'}[\boldsymbol{\Sigma}] = \Omega' = F[\boldsymbol{\Sigma}] - \mathrm{Tr}\ln(-\mathbf{G}') , \tag{108}$$

where $\Omega'$ is the cluster Hamiltonian's grand potential and $\mathbf{G}'$ its physical Green function, obtained through the exact solution. From this we can extract $F[\boldsymbol{\Sigma}]$ and it follows that

$$\Omega_{\mathbf{t}}[\boldsymbol{\Sigma}] = \Omega' + \mathrm{Tr}\ln(-\mathbf{G}') - \mathrm{Tr}\ln(-\mathbf{G}_{0\mathbf{t}}^{-1} + \boldsymbol{\Sigma}) = \Omega' + \mathrm{Tr}\ln(-\mathbf{G}') - \mathrm{Tr}\ln(-\mathbf{G}) \tag{109}$$

where $\mathbf{G}$ now stands for the CPT Green function (53). This expression can be further simplified as

$$\Omega_{\mathbf{t}}[\boldsymbol{\Sigma}] = \Omega' - \mathrm{Tr}\ln(\mathbf{1} - \mathbf{V}\mathbf{G}') \tag{110}$$

Let us finally make the trace more explicit: It is a sum over frequencies and a sum over lattice sites (and spin and band indices), which can be expressed instead as a sum over reduced wave-vectors (as the CPT Green function is diagonal in that index), plus a small trace (denoted tr) on residual indices (cluster site, spin, and band):

$$\Omega_{\mathbf{t}}[\boldsymbol{\Sigma}] = \Omega' + \int \frac{d\omega}{2\pi i}\frac{L}{N}\sum_{\tilde{\mathbf{k}}} \mathrm{tr}\ln\left[\mathbf{1} - \mathbf{V}(\tilde{\mathbf{k}})\mathbf{G}'(i\omega)\right] \tag{111}$$

$$= \Omega' + \int \frac{d\omega}{2\pi i}\frac{L}{N}\sum_{\tilde{\mathbf{k}}} \ln\det\left[\mathbf{1} - \mathbf{V}(\tilde{\mathbf{k}})\mathbf{G}'(i\omega)\right] \tag{112}$$

where the matrix identity $\text{tr} \ln A = \ln \det A$ was used in the second equation.

The type III approximation comes from the fact that the self-energy $\boldsymbol{\Sigma}$ is restricted to the exact self-energy of the cluster problem $H'$, so that variational parameters appear in the definition of the one-body part of $H'$. To come back to the question of the Weiss field $M$ introduced at the beginning of this section, we would set its value by solving the cluster Hamiltonian – i.e., computing $\Omega'$ and $\mathbf{G}'$ – for many different values of $M$ and evaluate the functional (111) for each of them, selecting the value that makes Expression (111) stationary. This is the idea behind the variational cluster approximation (VCA), described in more detail below.

In practice, we look for values of the cluster one-body parameters $\mathbf{t}'$ such that $\delta\Omega_\mathbf{t}/\delta\mathbf{t}' = 0$. It is useful for what follows to write the latter equation formally, although we do not use it in actual calculations. Given that $\Omega'$ is the actual grand potential evaluated for the cluster, $\partial\Omega'/\partial\mathbf{t}'$ is canceled by the explicit $\mathbf{t}'$ dependence of $\text{Tr} \ln(-\mathbf{G}_{0\mathbf{t}'}^{-1} + \boldsymbol{\Sigma})$ and we are left with

$$0 = \frac{\delta\Omega_\mathbf{t}[\boldsymbol{\Sigma}]}{\delta\boldsymbol{\Sigma}}\frac{\delta\boldsymbol{\Sigma}}{\delta\mathbf{t}'} = -\text{Tr}\left[\left(\frac{1}{\mathbf{G}_{0\mathbf{t}'}^{-1} - \boldsymbol{\Sigma}} - \frac{1}{\mathbf{G}_{0\mathbf{t}}^{-1} - \boldsymbol{\Sigma}}\right)\frac{\delta\boldsymbol{\Sigma}}{\delta\mathbf{t}'}\right] \tag{113}$$

or, more explicitly,

$$\sum_\omega \sum_{\mu\nu}\left[\left(\frac{1}{\mathbf{G}_{0\mathbf{t}'}^{-1} - \boldsymbol{\Sigma}(\omega)}\right)_{\mu\nu} - \frac{L}{N}\sum_{\tilde{\mathbf{k}}}\left(\frac{1}{\mathbf{G}_{0\mathbf{t}}^{-1}(\tilde{\mathbf{k}}) - \boldsymbol{\Sigma}(\omega)}\right)_{\mu\nu}\right]\frac{\delta\Sigma'_{\nu\mu}(\omega)}{\delta\mathbf{t}'} = 0. \tag{114}$$

where Greek indices are used for compound indices gathering cluster site, spin and possible band indices.

## 5.2 Variational Cluster Approximation

The Variational Cluster Approximation [19,26] (VCA), sometimes called Variational Cluster Perturbation Theory (VCPT), can be viewed as an extension of Cluster Perturbation Theory in which some parameters of the cluster Hamiltonian are set according to Potthoff's variational principle through a search for saddle points of the functional (111). The cluster Hamiltonian $H_c$ is typically augmented by Weiss fields, such as the Néel field (100) that allow for broken symmetries that would otherwise be impossible within a finite cluster. The hopping terms and chemical potential within $H_c$ may also be treated like additional variational parameters. In contrast with Mean-Field theory, these Weiss fields are not mean fields, in the sense that they do not coincide with the corresponding order parameters. The interaction part of $H$ (or $H_c$) is not factorized in any way and short-range correlations are treated exactly. In fact, the Hamiltonian $H$ is not altered in any way; the Weiss fields are introduced to let the variational principle act on a space of self-energies that includes the possibility of specific long-range orders, without imposing those orders.

Steps towards a VCA calculation are as follows:

1. Choose the Weiss fields to add, aided by intuition about the possible broken symmetries to expect.

2. Set up a procedure to calculate the functional (111).

3. Set up a procedure to optimize the functional, i.e., to find its saddle points, in the space of variational parameters.

4. Calculate the properties of the model at the saddle point.

## 5.3   Practical calculation of the Potthoff functional

Let $\mathbf{x}$ denote the (finite) set of variational parameters to be used. The Potthoff functional becomes the function

$$\Omega_{\mathbf{t}}(\mathbf{x}) = \Omega' - \int \frac{d\omega}{2\pi i} \frac{L}{N} \sum_{\tilde{\mathbf{k}}} \ln \det \left[ \mathbf{1} - \mathbf{V}(\tilde{\mathbf{k}}) \mathbf{G}'(\tilde{\mathbf{k}}, \omega) \right] \tag{115}$$

Once the cluster Green function is known by the methods described in Sect. 4, calculating the functional (115) requires an integral over frequencies and wave-vectors of an expression that requires a few linear-algebraic operations to evaluate. Two different methods have been used to compute these sums, described in what follows. The second method, entirely numerical, is much faster than the first one, which is partly analytic.

The integral over frequencies in (111) may be done analytically, with the result [27]

$$\Omega(\mathbf{x}) = \Omega'(\mathbf{x}) - \sum_{\omega_r' < 0} \omega_r' + \frac{L}{N} \sum_{\tilde{\mathbf{k}}} \sum_{\omega_r(\tilde{\mathbf{k}}) < 0} \omega_r(\tilde{\mathbf{k}}) \tag{116}$$

where the $\omega_r'$ are the poles of the Green function $\mathbf{G}'$ in the Lehmann representation (35) and the $\omega_r(\tilde{\mathbf{k}})$ are the poles of the VCA Green function $(\mathbf{G}_0^{-1}(\tilde{\mathbf{k}}) - \mathbf{\Sigma})^{-1}$. The latter are the eigenvalues of the $R \times R$ matrix $\mathbf{L}(\tilde{\mathbf{k}}) = \mathbf{\Lambda} + \mathbf{Q}^\dagger \mathbf{V}(\tilde{\mathbf{k}}) \mathbf{Q}$. $R$ is the number of columns of the Lehmann representation matrix $\mathbf{Q}$, basically the total number of iterations performed in the band Lanczos procedure.

In practice, the first sum in (116) is readily calculated. The second sum demands an integration over wave-vectors. For each wave-vector $\tilde{\mathbf{k}}$, one must calculate $\mathbf{L}(\tilde{\mathbf{k}})$ and find its eigenvalues, a process of order $R^3$. Other linear-algebraic manipulations leading to the diagonalization of $\mathbf{L}(\tilde{\mathbf{k}})$ are typically less time-consuming than the diagonalization itself. The computation time therefore goes like $N_k R^3$, where $N_k$ is the number of points in a mesh covering the reduced Brillouin zone (in fact half of the reduced Brillouin zone, since inversion symmetry is assumed).

In practice, it is therefore more efficient to compute frequency and momentum sums numerically. This is the approach followed in `pyqcm`, with the help of the external library `cuba` for multidimensional integrals, more specifically with a deterministic method (`Cuhre`). This approach avoids the diagonalization of medium-size matrices and the integration method being adaptive, only the frequencies close to the real axis require a high resolution momentum grid in some regions of the Brillouin zone.

## 5.4   Example : Antiferromagnetism

The Weiss field appropriate to Néel antiferromagnetism is defined in (100). Fig. 4 shows the Potthoff functional as a function of Néel Weiss field $M$ for various values of $U$, at half-filling, calculated on a $2 \times 2$ cluster. We note three solutions per curve: two equivalent minima located symmetrically about $M = 0$, and a maximum at $M = 0$ corresponding to the normal state solution. The normal and AF solutions both correspond to half-filling, and the AF solution has a lower energy density $\mathscr{E} = \Omega + \mu n$. We therefore conclude, on this basis, that the system has AF long-range order. Note that, as $U$ is increased, the profile of the curve is shallower and the minimum closer to zero. Indeed, for large $U$, the half-filled Hubbard model is well approximated by the Heisenberg model with exchange $J = 4t^2/U$, and the curve should (and will) scale towards a fixed shape when $\Omega/J$ is plotted against $M/J$ (both dimensionless quantities). Fig. 5 shows how the optimal Weiss field and the Néel order parameter vary as a function of $U$. The Weiss field vanishes both as $U \to 0$, where the order disappears, and as $U \to \infty$. In both limits the energy difference between

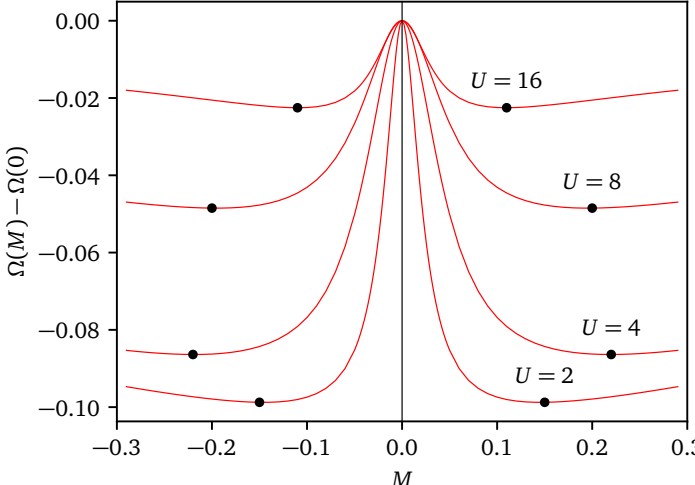

Figure 4: Potthoff functional as a function of Néel Weiss field $M$ for various values of $U$, at half-filling, calculated on a $2 \times 2$ cluster. The positions of the minima are indicated.

normal and broken symmetry state (or 'condensation energy') goes to zero (Fig. 5), and so should the critical (Néel) temperature. The order parameter $\bar{M} = \langle \hat{M} \rangle / N$ increases monotonically with $U$ and saturates.

Fig. 6 shows the Potthoff functional as a function of Néel Weiss field $M$ for various cluster sizes, at half-filling and $U = 8$. There is a clear and monotonous size dependence of the position of the minimum. In particular, the optimal Weiss field decreases as cluster size increases. This should not worry us, quite on the contrary. The Weiss field is needed only because spontaneously broken symmetries cannot arise on a finite cluster. The bigger the cluster, the easier it is to break the symmetry and the optimal Weiss field should tends towards zero as the cluster size goes to infinity. Finite-size scaling is generally very difficult, because cluster sizes are small and clusters vary in shape as well as size. Moreover, open boundary conditions are used rather than periodic ones, which adds edge effects to size effects. One needs to define a scaling parameter $q$, ranging between 0 and 1, that somehow defines the "quality" of the cluster ($q = 1$ being the thermodynamic limit). Fig. 7 shows the optimal Néel Weiss field as a function of two possibilities for the scaling factor $q$, for the half-filled Hubbard model at $U = 16$. The first possibility (blue dots) is $q = 1 - 1/L$, which does not take into account the shape of the cluster. The second possibility (red dots) corresponds to $q$ defined as the number of links on the cluster, divided by twice the number of sites. This also goes to 1 in the thermodynamic limit (for the square lattice), but this time takes into account the boundary of the cluster. Indeed, $1 - q$ corresponds to the fraction of links of the lattice that are "inter-cluster" and thus treated "perturbatively" in the CPT sense. In that case, the scaling is good, as the optimal Weiss fields extrapolates very close to zero in the $q \to 1$ limit. At the same time, the AF order parameter also decreases, but extrapolates to a finite value, as shown on the same figure

## 5.5   Superconductivity

Superconductivity requires the use of pairing fields as Weiss fields, i.e., of operators creating Cooper pairs at specific locations. Generally, pairing fields have the form

$$\hat{\Delta} = \sum_{\mathbf{rr}'} \Delta_{\mathbf{rr}'} c_{\mathbf{r}\uparrow} c_{\mathbf{r}'\downarrow} + \text{H.c} \tag{117}$$

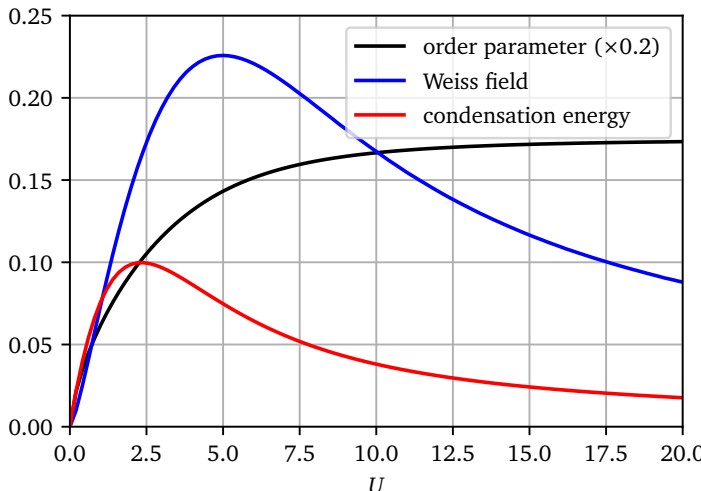

Figure 5: Optimal Néel Weiss field $M$ and corresponding order parameter, as a function of $U$, at half-filling, calculated on a $2 \times 2$ cluster. Also shown is the ordering energy, i.e., the difference between the energy density of the normal state and that of the Néel state (in fact the difference between the grand potentials of the two solutions, since they both sit at half-filling).

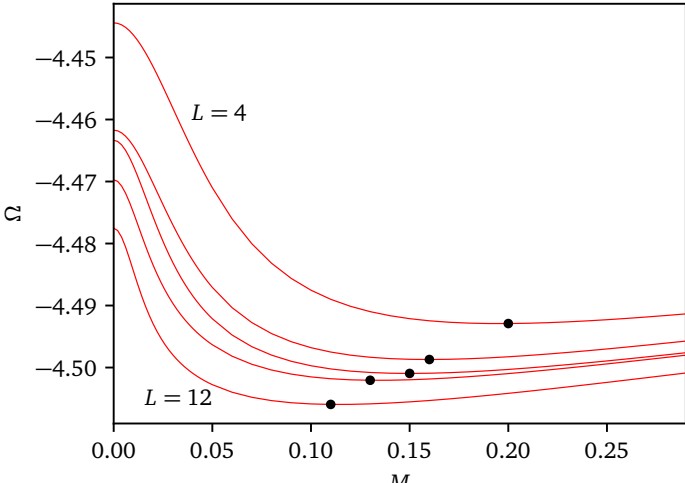

Figure 6: Potthoff functional as a function of Néel Weiss field $M$ for various cluster sizes, at half-filling and $U = 8$. The clusters used (from top to bottom) are: $2 \times 2$, $2 \times 3$, $2 \times 4$, B10 – see Fig. 1 –, and $3 \times 4$. The positions of the minima are indicated.

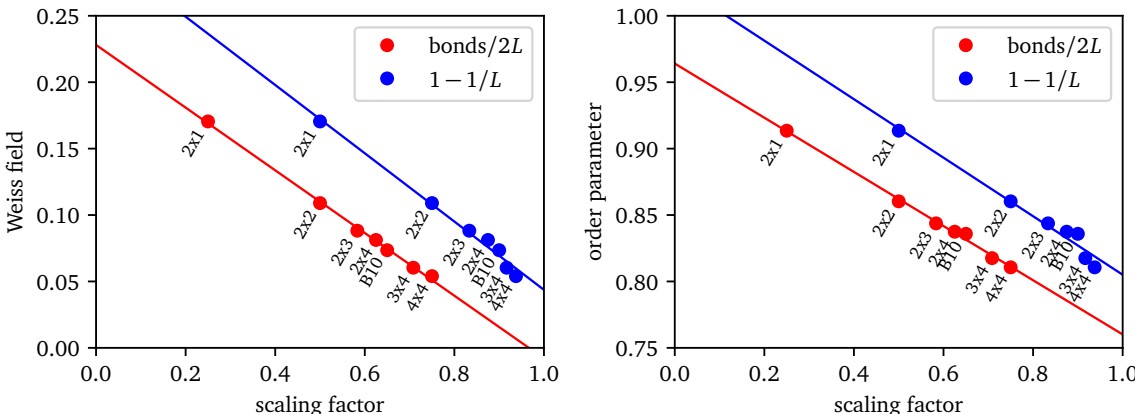

Figure 7: Top: Optimal Néel Weiss field for the half-filled Hubbard model at $U = 16$, as a function of scaling parameter. Blue points : the scaling parameter is $1 - 1/L$, and the scaling is poor. Red points : the scaling parameters is the number of cluster links divided by $2L$ – this takes open boundary conditions into account. We see how the Weiss field goes to zero in the thermodynamic limit. Bottom: Same, for the Néel order parameter, which tends to a finite value in the thermodynamic limit. Against the second scaling parameter works better.

Different types of superconductivity correspond to different pairing functions $\Delta_{\mathbf{rr'}}$. For instance, ordinary (local) $s$-wave pairing (*à la* BCS) corresponds to $\Delta_{\mathbf{rr'}} = \delta_{\mathbf{rr'}}$. On a square lattice, what is usually known as $d_{x^2-y^2}$ pairing corresponds to

$$\Delta_{\mathbf{rr'}} = \begin{cases} 1 & \text{if } \mathbf{r} - \mathbf{r}' = \pm\mathbf{e}_x \\ -1 & \text{if } \mathbf{r} - \mathbf{r}' = \pm\mathbf{e}_y \end{cases} \tag{118}$$

whereas $d_{xy}$ pairing corresponds to

$$\Delta_{\mathbf{rr'}} = \begin{cases} 1 & \text{if } \mathbf{r} - \mathbf{r}' = \pm(\mathbf{e}_x + \mathbf{e}_y) \\ -1 & \text{if } \mathbf{r} - \mathbf{r}' = \pm(\mathbf{e}_x - \mathbf{e}_y) \end{cases} \tag{119}$$

($\mathbf{e}_{x,y}$ are lattice vectors on the square lattice). The above two pairing are spin singlets.

Pairing fields, once introduced in the cluster Hamiltonian $H_c$ as Weiss fields, do not conserve particle number (but conserve spin). This increases the computational burden, since now the Hilbert space must be increased to include all sectors of a given total spin. In practice, one uses the Nambu formalism, which in this case amounts to a particle-hole transformation for spin-down operators. Indeed, if we introduce the operators

$$c_{\mathbf{r}} = c_{\mathbf{r}\uparrow} \quad \text{and} \quad d_{\mathbf{r}} = c_{\mathbf{r}\downarrow}^{\dagger} \tag{120}$$

then the pairing fields look like simple hopping terms between $c$ and $d$ electrons, and the whole cluster Hamiltonian can be kept in the standard form (1), albeit with hybridization between $c$ and $d$ orbitals.

Fig. 8 illustrates the dependence of the Potthoff functional on various superconducting pairing fields (generically denoted $\Delta$). In that case, only $d_{x^2-y^2}$ pairing leads to a nontrivial solution. Others are piece-wise monotonously increasing or decreasing function, with a single zero-derivative point at $\Delta = 0$.

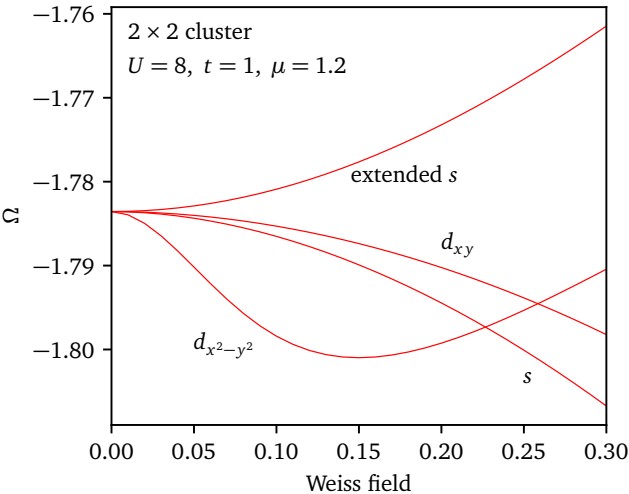

Figure 8: Profile of the Potthoff functional as a function of Weiss field for various super-conducting pairing fields. The extended $s$-wave is defined as the same as in (118), but without the sign change between $x$ and $y$ directions.

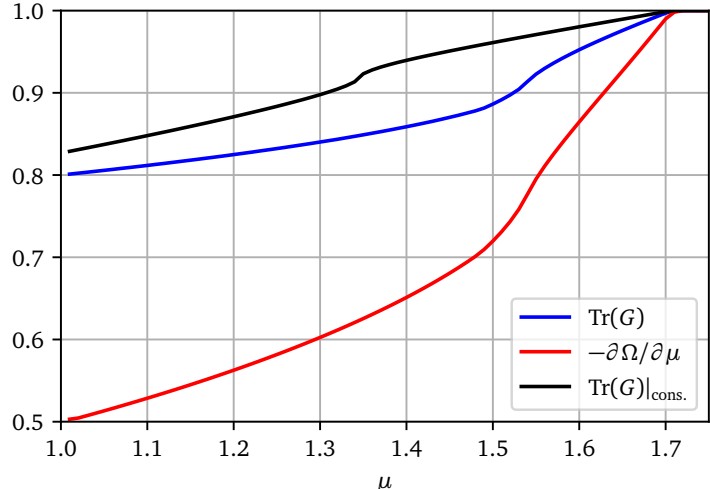

Figure 9: Comparisons of the estimates of the electron density $n$ as a function of chemical potential $\mu$, with different methods of calculation, for the normal solution, at $U = 8$, on a $2 \times 2$ cluster. The subscript 'cons.' means that the corresponding quantities were computing in a thermodynamically consistent way, by using $\mu'$ as a variational parameter.

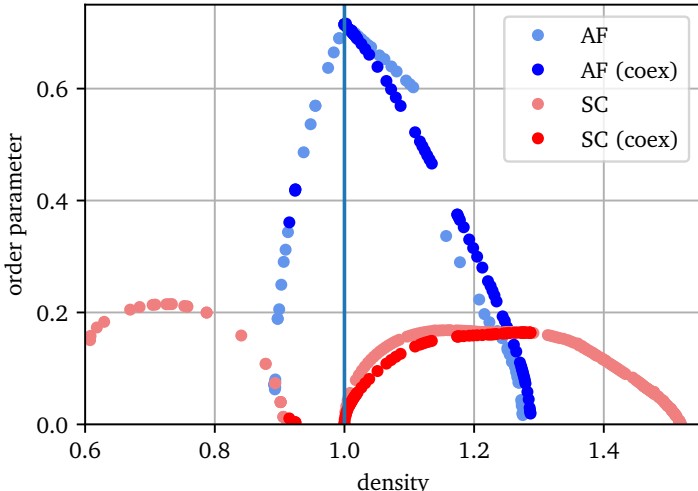

Figure 10: Order parameters for $d_{x^2-y^2}$ pairing and Néel antiferromagnetism for a model of the high-$T_c$ cuprates with $U = 8$, diagonal hopping $t_1 = -0.3$ and third neighbor hopping $t_2 = 0.2$. Calculations are performed on a $3 \times 4$ cluster. Three solutions are displayed: (1) a pure $d_{x^2-y^2}$, obtained with two variational parameters ($\mu'$ and $\Delta_{x^2-y^2}$); (2) a pure Néel solution obtained by varying $\mu'$ and the Néel Weiss field $M$; a homogeneous coexistence solution obtained by varying $\mu'$, $M$ and $\Delta_{x^2-y^2}$ (data from Ref. [28]).

## 5.6   Thermodynamic consistency

One of the main difficulties associated with VCA (or CPT) is the limited control over electron density. In the absence of pairing fields, electron number is conserved and clusters have a well-defined number of electrons. This makes a continuously varying electron density a bit hard to represent. Of course, one may simply vary the chemical potential $\mu$ and look at the corresponding variation of the electron density, given by the functional trace of the Green function $\mathrm{Tr}\,\mathbf{G}$ (see Eq. 52). This provides a continuously varying estimate of the density as a function of $\mu$. An alternate way of estimating the density is to use the relation

$$n = -\frac{\partial \Omega}{\partial \mu} \tag{121}$$

where the grand potential $\Omega$ is approximated by the Potthoff functional at the solution found, and $\mu$ is varied as an external parameter. The problem is that the two estimates do not coincide (see Fig. 9). In other words, the approach is not thermodynamically consistent. The recipe to make it consistent is simple : the chemical potential $\mu'$ of the cluster should not be assumed to be the same as that of the lattice system ($\mu$), but should be treated as a variational parameter. If this is done, then the two methods for computing $n$ given precisely the same result (see Fig. 9), and this can easily be proven in general. Results on a Hubbard model for the cuprates with thermodynamic consistency are shown on Fig. 10; see also Ref. [27].

## 5.7   Searching for stationary points

Let $x_i$ be the $n$ different variational parameters used in VCA, making up the array $\mathbf{x}$. Once the function $\Omega(\mathbf{x})$ may be efficiently calculated, it remains to find a stationary point of that function. This point is not necessarily a minimum in all directions. Indeed, experience has shown that $\omega$ is a maximum as a function of the cluster chemical potential $\mu'$, while it is generally a minimum as a function of symmetry-breaking Weiss fields like $M$ or $\Delta$.

The Newton-Raphson algorithm allows one to find stationary points with a small number of function evaluations. One starts with a trial point $\mathbf{x}_0$ and an initial step $h$. Let $\mathbf{e}_i$ denote the unit vector in the direction of axis $i$ of the variational space. The function $\omega$ is then calculated at as many points as necessary to fit a quadratic form in the neighborhood of $\mathbf{x}_0$. This requires $(n+1)(n+2)/2$ evaluations, at points like $\mathbf{x}_0$, $\mathbf{x}_0 \pm h\mathbf{e}_i$, and a few of $\mathbf{x}_0 + h(\mathbf{e}_i + \mathbf{e}_j)$. The stationary point $\mathbf{x}_1$ of that quadratic form is then used as a new starting point, the step $h$ is reduced to a fraction of the difference $|\mathbf{x}_1 - \mathbf{x}_0|$, and the process is iterated until convergence on $|\mathbf{x}_i - \mathbf{x}_{i-1}|$ is achieved. A variant of this method, the quasi-Newton algorithm, may also be used, in which the full Hessian matrix of second derivatives is not calculated. It requires in general more iterations, but fewer function evaluations at each step.

The advantage of the Newton-Raphson method lies in its economy of function evaluations, which are very expensive here: each requires the solution of the cluster Hamiltonian. Its disadvantage is a lack of robustness. One has to be relatively close to the solution in order to converge towards it. But one typically runs parametric studies in which an external (i.e. non variational) parameter of the model is varied, such as the chemical potential $\mu$ or the interaction strength $U$. In this context, the solution associated with the current value of the external parameter may be used as the starting point for the next value, and in this fashion, by proximity, one may conduct rather robust calculations.

A more robust method, albeit more time consuming, is the conjugate-gradient algorithm, which we will not explain here as it is amply documented and fairly common. However, this algorithm finds minima (or maxima), not saddle points in general. We must therefore take the extrinsic step of identifying parameters (like $\mu'$ above) that are expected to drive maxima of $\omega$, and a complementary set of parameters (like $M$ and $\Delta$ above) that drive minima of $\omega$. One then, iteratively, finds maxima and minima with the two sets of parameters in succession, and stops when convergence on $|\mathbf{x}_i - \mathbf{x}_{i-1}|$ has been achieved. This method is suitable to find a first solution when the Newton-Raphson method fails to deliver one. It may however converge to minima that are in fact singularities of $\omega$, i.e., points where the derivatives are not defined. Such points may occur as the result of energy-level crossings in clusters and are an artifact of the finite-cluster size.

# 6 The Cellular Dynamical Mean Field Theory

The Cellular dynamical mean-field theory (CDMFT) – also called Cluster dynamical mean-field theory – is a cluster extension of Dynamical mean-field theory (DMFT). Since there is no real pedagogical gain in describing first DMFT, we will proceed directly to CDMFT, in the context of a an exact diagonalization solver.

The basic idea behind CDMFT is to approximate the effect on the cluster of the remaining degrees of freedom of the lattice by a bath of uncorrelated orbitals that exchange electrons with the cluster, and whose parameters are set in a self-consistent way. Explicitly, the cluster Hamiltonian $H_c$ takes the form

$$
\begin{aligned}
H_c = & -\sum_{\mu,\nu} t_{\mu\nu} c_\mu^\dagger c_\nu + U \sum_{\mathbf{R}} n_{\mathbf{R}\uparrow} n_{\mathbf{R}\downarrow} \\
& + \sum_{\mu,\alpha} \theta_{\mu\alpha} (c_\mu^\dagger a_\alpha + \text{H.c.}) + \sum_\alpha \varepsilon_\alpha a_\alpha^\dagger a_\alpha
\end{aligned}
\tag{122}
$$

where $a_\alpha$ annihilates an electron on a bath orbital labeled $\alpha$. The label $\alpha$ includes both an 'bath site' index and a spin index for that 'site'. The bath is characterized by the energy of each orbital ($\varepsilon_\alpha$) and by the bath-cluster hybridization matrix $\theta_{\mu\alpha}$ (the index $\mu$ includes

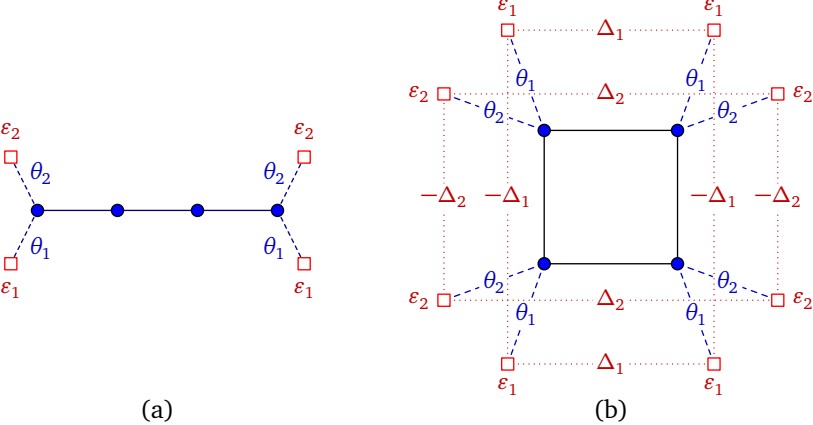

Figure 11: Examples of clusters with baths for use in CDMFT. Bath sites are square, cluster sites blue circles. Bath energies $\varepsilon_i$, hybridizations $\theta_i$ are indicated. System (a) is appropriate for studying the one-dimensional Hubbard model and CDMFT results are shown on Fig. 12. System (b) is appropriate for the two-dimensional Hubbard model (in-bath pairing operators $\Delta_i$ are shown) and CDMFT results are shown on Fig. 14.

cluster site, spin and band indices). This representation of the environment through an Anderson impurity model was introduced in Ref. [29] in the context of DMFT (i.e., a single-site cluster). Note that 'bath site' is a misnomer, as bath orbitals have no position assigned to them. Because of the analogy with the Anderson impurity model (AIM), the cluster-bath system is often referred to as the *impurity* (even though no disorder is involved) and the method used to compute the cluster Green function is called the *impurity solver*.

The effect of the bath on the electron Green function is encapsulated in the so-called hybridization function

$$\Gamma_{\mu\nu}(\omega) = \sum_{\alpha} \frac{\theta_{\mu\alpha}\theta_{\nu\alpha}^*}{\omega - \varepsilon_\alpha} \tag{123}$$

which enters the electron Green function as

$$\mathbf{G}_c^{-1} = \omega - \mathbf{t} - \mathbf{\Gamma}(\omega) - \mathbf{\Sigma}_c(\omega) \tag{124}$$

By definition, the only effect of adding the electron-electron interaction is to add the self-energy $\mathbf{\Sigma}_c$, as above.

Note that while the CPT relation (55) is still valid, the relation (54) must be modified in the presence of a bath in order to compensate for the hybridization function:

$$\mathbf{G}^{-1}(\tilde{\mathbf{k}}, \omega) = \mathbf{G}_c^{-1}(\omega) + \mathbf{\Gamma}(\omega) - \mathbf{V}(\tilde{\mathbf{k}}) \ . \tag{125}$$

## 6.1 Bath degrees of freedom and SFA

The CDMFT Hamiltonian (122) defines a valid reference system for Potthoff's self-energy functional approach, since it shares the same interaction part as the lattice Hamiltonian $H$ and since each cluster of the super-lattice has its own identical, independent copy. From the SFA point of view, the bath parameters $\{\varepsilon_\alpha, \theta_{\mu\alpha}\}$ can in principle be chosen in such a way as to make the Potthoff functional stationary. A subtlety arises: the bath system must be considered part of the original Hamiltonian $H$, albeit without hybridization to the cluster sites, in order for both Hamiltonians to describe the same degrees of freedom;

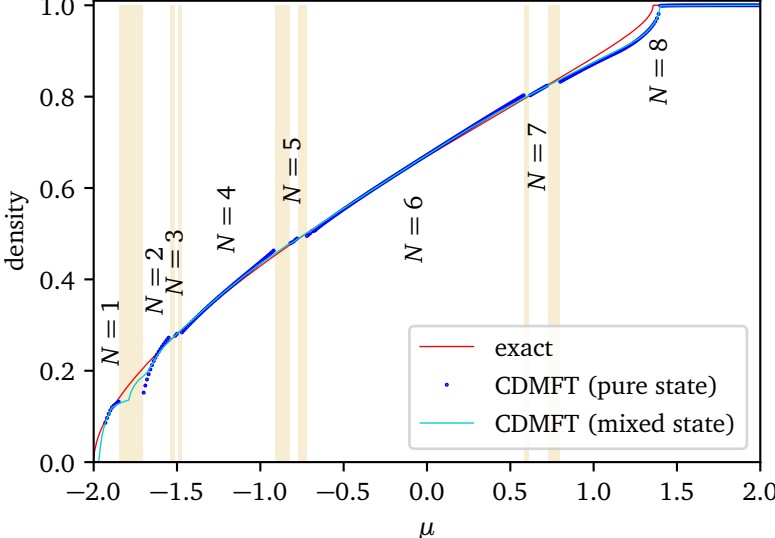

Figure 12: Electron density as a function of chemical potential, for the one-dimensional Hubbard model, at $U = 4$. The red curve is the exact result from the Lieb-Wu solution using the Bethe Ansatz. The CDMFT results are obtained from the 4-site cluster of Fig. 11(a). The blue dots are obtained when allowing only pure states (well defined electron number on the impurity system, as labeled). The shaded areas show intervals in $\mu$ where such pure solutions cannot be found. The turquoise line is obtained by allowing mixed states for the impurity, with a temperature $T = 0.01$ (in units of the hopping $t$).

but within $H$ we are free to give the bath trivial parameters ($\varepsilon_\alpha = 0$). Performing VCA-like calculations with bath degrees of freedom is possible, but difficult and in practice restricted to simple systems [30–32].

When evaluating the Potthoff functional in the presence of a bath, one must add a contribution from the bath to $\mathrm{Tr}\ln(-\mathbf{G}_c)$, which takes the form

$$\Omega_{\mathrm{bath}} = \sum_{\varepsilon_\alpha < 0} \varepsilon_\alpha \qquad (126)$$

and which comes from the zeros of the cluster Green function induced by the poles of the hybridization function. Note that the zeros coming from the self-energy cancel out in Eq. (116) between the contribution of $\mathrm{Tr}\ln(-\mathbf{G}_c)$ and that of $\mathrm{Tr}\ln(-\mathbf{G})$, but not those coming from $\mathbf{\Gamma}(\omega)$, as they only occur in $\mathbf{G}_c$.

## 6.2   The CDMFT self-consistent procedure

In practice, CDMFT does not look for a strict solution of the Euler equation (114). It tries instead to set each of the terms between brackets to zero separately. Since the Euler equation (114) can be seen as a scalar product, CDMFT requires that the modulus of one of the vectors vanish to make the scalar product vanish. From a heuristic point of view, it is as if each component of the Green function in the cluster were equal to the corresponding component deduced from the lattice Green function. Clearly, the left-hand side of Eq. (114) cannot vanish separately for each frequency, since the number of degrees of freedom in the bath is insufficient. Instead, one adopts the following self-consistent scheme (see Fig. 13):

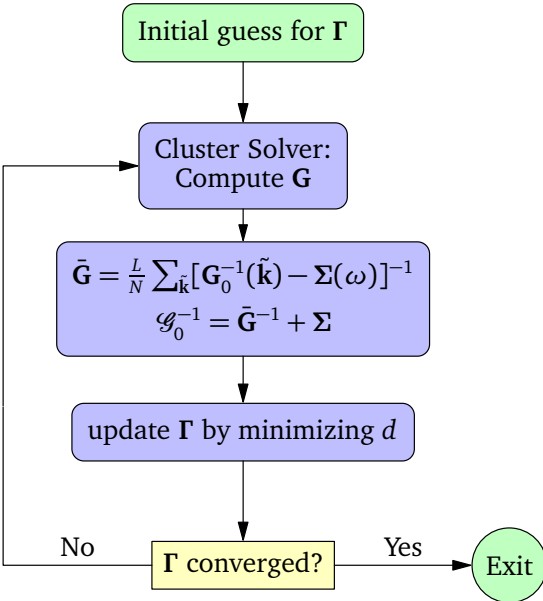

Figure 13: The CDMFT algorithm with an exact diagonalization solver.

1. Start with a guess value of the bath parameters $(\theta_{\mu\alpha}, \varepsilon_{\alpha})$, that define the hybridization function (123).

2. Solve for the cluster Green function $\mathbf{G}(\omega)$ with the impurity solver (here ED).

3. Calculate the super-lattice-averaged Green function

$$\bar{\mathbf{G}}(\omega) = \frac{L}{N} \sum_{\tilde{\mathbf{k}}} \frac{1}{\mathbf{G}_0^{-1}(\tilde{\mathbf{k}}) - \mathbf{\Sigma}_c(\omega)} \tag{127}$$

and the combination

$$\mathscr{G}_0^{-1}(\omega) = \bar{\mathbf{G}}^{-1} + \mathbf{\Sigma}_c(\omega) \tag{128}$$

4. Minimize the following distance function:

$$\begin{aligned}
d &= \sum_{i\omega_n,\nu,\nu'} W_n \left| \left( \mathbf{G}(\omega)^{-1} - \bar{\mathbf{G}}(\omega)^{-1} \right)_{\nu\nu'} \right|^2 \\
&= \sum_{i\omega_n,\nu,\nu'} W_n \left| \left( i\omega_n + \mu - \mathbf{t}_c - \mathbf{\Gamma}(i\omega_n) - \mathscr{G}_0^{-1} \right)_{\nu\nu'} \right|^2
\end{aligned} \tag{129}$$

over the set of bath parameters. Changing the bath parameters at this step does not require a new solution of the Hamiltonian $H_c$, but merely a recalculation of the hybridization function $\mathbf{\Gamma}$ (123). The weights $W_n$ are chosen arbitrarily but with common sense.

5. Go back to step (2) with the new bath parameters obtained from this minimization, until they are converged.

In practice, the distance function (129) can take various forms, for instance by choosing frequency-dependent weights $W_n$ in order to emphasize low-frequency properties [13,33,34] or by using a sharp frequency cutoff [35]. These weights $W_n$ can be considered as rough approximations for the missing factor $\delta\Sigma'_{\nu\mu}(\omega)/\delta\mathbf{t}_c$ in the Euler equation (114). The

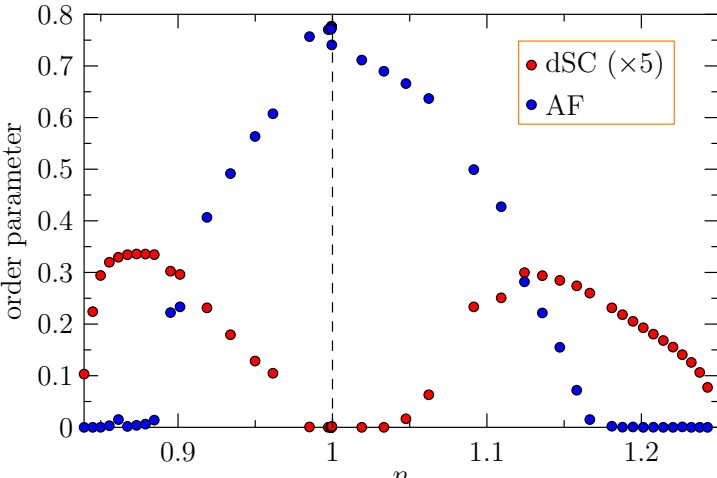

Figure 14: Néel (AF) and *d*-wave (dSC) order parameters obtained from CDMFT applied to the (4+8)-cluster of Fig. 11(c), for the two-dimensional Hubbard model with $U = 8$ and diagonal hopping $t' = -0.3$. The data is shown as a function of the calculated lattice density $n$. The order parameters are calculated using the same operators as in the corresponding VCA calculation illustrated on Fig. 8, even though these operators played no role in the solution: they are merely used as a probe. In this calculation, we set $\beta = 20$ and a sharp cutoff $\omega_c = 3$ was used. The dSC and AF solutions were both allowed simultaneously (9 bath parameters) and there are regions of coexistence of the two orders.

frequencies are summed over on a discrete, regular grid along the imaginary axis, defined by some fictitious inverse temperature $\beta$, typically of the order of 50 (in units of $t^{-1}$). Even when the total number of cluster plus bath sites in CDMFT equals the number of sites in a VCA calculation, CDMFT is much faster than the VCA since the minimization of a grand potential functional requires many exact diagonalizations of the cluster Hamiltonian $H_c$.

## 6.3   Examples

Let us start with a one-dimensional example. Fig. 12 shows the electron density $n$ as a function of the chemical potential $\mu$ for the one-dimensional Hubbard model, as computed from CDMFT with the system illustrated in Fig. 11(a). The red curve is the exact result from the Lieb-Wu solution [36]. The blue dots are the CDMFT solutions obtained by imposing a pure state for the impurity, i.e., with a definite number of electrons on the cluster-bath system, from $N = 8$ down to $N = 1$. Note that even though $N$ is fixed on the impurity, it is flexible on the cluster per se because of the presence of the bath orbitals. In fact, states with an odd number of electrons are not pure, since the ground state is degenerate between two states with $S_z = \pm\frac{1}{2}$, and these two states (and the corresponding Green functions) are computed separately. Note that there are intervals of $\mu$ (shaded in yellow) where CDMFT finds no consistent ground state. By that we mean that the CDMFT procedure done within a specific value of $N$ may converge to a set of bath parameters, but the lowest-energy state in that Hilbert space sector is not the true ground state, which would reside in a sector with a different value of $N$. On the other hand, if we allow mixed states between different sectors, with a small temperature $T$ (here $T = 0.01t$), then a solution is found (turquoise curve) for all values of $\mu$ that interpolates well between the solutions found with a fixed value of $N$ on the impurity.

Next, consider the two-dimensional cluster illustrated in Fig. 11(c). This 4-site, 8-

bath site cluster is the main cluster used in CDMFT simulations of high-$T_c$ cuprates using the two-dimensional Hubbard model. It is useful in that case to view the orbitals numbered 5 to 8 as a first bath set, and the orbitals numbered 9 to 12 as a second bath. Each site of the cluster is connected to one orbital of each set. In studying the normal state, and taking into account the symmetries of the cluster, we would need 4 bath parameters: one bath-cluster hopping and one bath energy for each set. In order to treat a possible antiferromagnetic phase, one must modify the bath energies and hopping in a spin-dependent way. The gray and white squares on the figure then distinguish orbitals of a given bath according to their shift in site energy (of opposite signs for opposite spins). The corresponding bath-cluster hybridization may also be different, which makes a total of 8 parameters. Finally, in order to study $d$-wave superconductivity, we introduce pairing within each bath (red dotted lines on the figure), vertical and horizontal pairing being of opposite signs. This introduces an additional parameter, for a total of 9. At this point, an important remark is in order : Formula (123) for the hybridization function only applies if the bath orbitals are not hybridized between themselves. The $d$-wave pairing just described certainly breaks that condition. This is not a problem, however, if we perform a change of variables within bath degrees of freedom (a Bogoliubov transformation) prior to solving the problem numerically, such as to make the bath Hamiltonian diagonal. Then the poles of the hybridization function no longer correspond to the bath energies as defined originally in the model, but rather to the eigenvalues of the bath Hamiltonian.

Results of a CDMFT calculation on this system are shown in Fig. 14. Comparing with the VCA result of Fig. 10, we notice first the similarities: the existence of a dSC phase away from half-filling for both electron and hole doping and the possibility of homogeneous coexistence between antiferromagnetism and $d$-wave superconductivity. But differences are obvious : the VCA diagram is more asymmetric than the CDMFT one in terms of electron vs hole doping. Both calculations agree on the critical doping for antiferromagnetism on the hole-doped side ($\sim 10\%$), but not on the electron-doped side. The VCA result does not show homogeneous coexistence between AF and dSC on the hole-doped side – although it appears on smaller clusters. In fact, the presence of homogeneous coexistence on the hole-doped side in CDMFT depends on the bath configuration; it appears in the simple bath configuration of Fig. 11(c), but not in a more general bath configuration [37].

# 7 Extended interactions

The methods described above (CPT, VCA, CDMFT) only apply to systems with on-site interactions, since the Hamiltonians $H$ and $H_c$ must differ by one-body terms only, i.e., they must have the same interaction part. If extended interactions are present, they are partially truncated when the lattice is tiled into clusters and one must apply further approximations. Specifically, the Hartree (or mean-field) decomposition can be applied on the extended interactions that straddle different clusters, while interactions (local or extended) within each cluster are treated exactly. This is called the dynamical Hartree approximation (DHA) and has been used in Ref [38] to study charge order in the extended, one-band Hubbard model and in Refs [39–41] in order to assess the effect of extended interactions on strongly-correlated or charge order. (the qualifier *dynamical* is used to reflect the presence of short-range correlations within the method and its association with methods based on the self-energy, such as VCA or CDMFT). We will explain this approach in this section.

Let us write Hamiltonian with extended interactions as

$$H = H_0(\mathbf{t}) + H_\text{ext} \qquad H_\text{ext} = \frac{1}{2} \sum_{i,j} V_{ij} n_i n_j \qquad (130)$$

where $i, j$ are compound indices for lattice site and orbital label, $n_{i\sigma}$ is the number of electrons of spin $\sigma$ on site/orbital $i$, $n_i = n_{i\uparrow} + n_{i\downarrow}$ and $H_0$ is the rest of the Hamiltonian, that could also contain on-site interactions or any interaction that does not straddle clusters. In the dynamical Hartree approximation, $H_\text{ext}$ in (130) is replaced by

$$H_\text{ext}^\text{DHA} = \frac{1}{2} \sum_{i,j} V_{ij}^\text{c} n_i n_j + \frac{1}{2} \sum_{i,j} V_{ij}^\text{ic} (\bar{n}_i n_j + n_i \bar{n}_j - \bar{n}_i \bar{n}_j) \qquad (131)$$

where $V_{ij}^\text{c}$ denotes the extended interaction between orbitals belonging to the *same* cluster, whereas $V_{ij}^\text{ic}$ those interactions between orbitals of *different* clusters. Here $\bar{n}_i$ is a mean-field, presumably the average of $n_i$, but not necessarily, as we will see below.

Let us express the index $i$ as a cluster index $c$ and a site-within-cluster index $\alpha$. Then Eq. (131) can be expressed as

$$\frac{1}{2} \sum_{c,\alpha,\beta} \tilde{V}_{\alpha\beta}^\text{c} n_{c,\alpha} n_{c,\beta} + \frac{1}{2} \sum_{c,\alpha,\beta} \tilde{V}_{\alpha\beta}^\text{ic} (\bar{n}_\alpha n_{c,\beta} + n_{c,\alpha} \bar{n}_\beta - \bar{n}_\alpha \bar{n}_\beta) \qquad (132)$$

where we have assumed that the mean fields $\bar{n}_i$ are the same on all clusters, i.e., they have minimally the periodicity of the super-lattice, hence $\bar{n}_i = \bar{n}_\alpha$. We have consequently replaced the large, $N \times N$ and block-diagonal matrix $V_{ij}^\text{c}$ by a small, $N_c \times N_c$ matrix $\tilde{V}_{\alpha\beta}^\text{c}$, and we have likewise "folded" the large $N \times N$ matrix $V_{ij}^\text{ic}$ into the $N_c \times N_c$ matrix $\tilde{V}_{\alpha\beta}^\text{ic}$.

To clarify this last point, consider the simple example of a one-dimensional lattice with nearest-neighbor interaction $v$, tiled with 3-site clusters. Then

$$H_\text{ext} = v \sum_{i=0}^{N} n_i n_{i+1} \qquad (133)$$

leads to the following $3 \times 3$ interaction matrices:

$$\tilde{V}^\text{c} = v \begin{pmatrix} 0 & 1 & 0 \\ 1 & 0 & 1 \\ 0 & 1 & 0 \end{pmatrix} \qquad \tilde{V}^\text{ic} = v \begin{pmatrix} 0 & 0 & 1 \\ 0 & 0 & 0 \\ 1 & 0 & 0 \end{pmatrix} \qquad (134)$$

In practice, the symmetric matrix $\tilde{V}_{\alpha\beta}^\text{ic}$ is diagonalized and the mean-field inter-cluster interaction is expressed in terms of eigen-operators $m_\mu$:

$$\hat{V}^\text{ic} = \sum_\mu D_\mu \left[ \bar{m}_\mu m_\mu - \frac{1}{2} \bar{m}_\mu^2 \right] \qquad (135)$$

For instance, in the above simple one-dimensional problem, these eigen-operators $m_\mu$ and their corresponding eigenvalues $D_\mu$ are

$$\begin{aligned} D_1 &= -v & m_1 &= (n_1 - n_3)/\sqrt{2} \\ D_2 &= \phantom{-}0 & m_2 &= n_2 \\ D_3 &= \phantom{-}v & m_3 &= (n_1 + n_3)/\sqrt{2} \end{aligned} \qquad (136)$$

($n_{1,2,3}$ are the electron number operators on each of the three sites of the cluster). The mean fields $\bar{n}_i$ are determined either by applying (i) self-consistency or (ii) a variational method. In the case of ordinary mean-field theory, in which the mean-field Hamiltonian is entirely free of interactions, these two approaches are identical. In the present case, where the mean-field Hamiltonian also contains interactions treated exactly within a cluster, self-consistency does not necessarily yield the same solution as energy minimization. In the first case, the assignation $\bar{n}_i \leftarrow \langle n_i \rangle$ would be used to iteratively improve on the value of $\bar{n}_i$ until convergence. In the second case, one could treat $\bar{n}_i$ like any other Weiss field in the VCA approach, except that $\bar{n}_i$ is not defined only on the cluster, but on the whole lattice. We will see in Sect. 8.9 how this is done in practice in the pyqcm library.

# 8    The PyQCM library

## 8.1    Access and general architecture

The `pyqcm` library is available on bitbucket.org. It contains a core written in `C++`, that compiles into a shared object library `qcm.so`. That is in turn included in a Python module called `pyqcm`, which contains submodules dedicated to CDMFT and VCA. The user does not have to interact with the shared object library `qcm.so` directly. Instructions for installations can be found in the repository, but it can be as simple as cloning the git repository and typing `pip install .` within the main source directory.

**Dependencies**   The library uses Lapack (or equivalent) for basic linear algebra. It uses the cuba library for multidimensional integrals. It optionally uses the eigen template library for representing the sparse Hamiltonian. `pyqcm` has its own efficient Lanczos, band Lanczos and Davidson-Liu methods coded in.

**Documentation**   The library's documentation can be produced by going to the distribution's `docs/` folder and issuing the command `./makedoc`. It is produced by Sphinx and is also available online on readthedocs. In the remainder of this section we provide a general introduction to the library, with examples, but without going into all the details of each functionality, which would take excessive time and space. We refer the reader to the complete documentation for that.

## 8.2    Defining models I : Geometry

In `pyqcm` , one defines a *lattice model* (in dimension 0 to 3), and one or more *cluster models*, the latter defining the impurity, i.e., the part of the model that is solved by exact diagonalization. The lattice model defines how the clusters are arranged to tile the infinite lattice, and contains lattice operators that are then restricted to the clusters and contribute to the cluster Hamiltonian. Let us illustrate this by two examples, one extremely rudimentary, and the second one a bit more sophisticated.

Consider the Hubbard Hamiltonian in dimension 1, which we decide to tile with identical clusters of size 4, a illustrated below (inter-cluster hopping terms are represented by dashed lines).

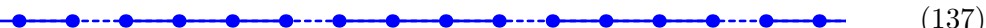

(137)

To define a basic nearest-neighbor hopping and a Hubbard $U$, the following simple code is required:

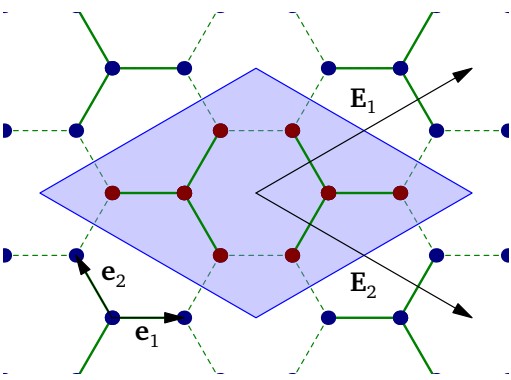

Figure 15

```
1  import pyqcm
2  CM = pyqcm.cluster_model(4)
3  clus = pyqcm.cluster(CM, ((0,0,0), (1,0,0), (2,0,0), (3,0,0)))
4  model = pyqcm.lattice_model('1D_4', clus, ((4,0,0),))
5  model.interaction_operator('U')
6  model.hopping_operator('t', (1,0,0), -1)
```

Line 2 initiates a cluster model containing 4 physical sites and no bath site, here stored in the object `CM`. Objects of type `cluster_model` have no notion of geometry or position. Line 3 defines a physical (geometric) cluster named `clus` with positions $(i, 0, 0)$ $(i = 0, \ldots, 3)$, based on the abstract cluster model `CM`. There could be more than one cluster based on the same model in the repeated unit, hence the distinction between the two objects. All positions are integer-component three-vectors, even for models in dimension $< 3$. Line 4 defines an object of type `lattice_model` named `model` with a super-lattice vector `(4,0,0)` and based on the unique cluster `clus` defined the line before. The lattice model is given then name `'1D_4'` that is used to refer to it in output files. Line 5 defines a local interaction operator named `U` and line 6 a nearest-neighbor hopping operator name `t`, with hopping vector $(1, 0, 0)$ and amplitude multiplier $-1$, so that the lattice Hamiltonian reads

$$H = -t \sum_{i,\sigma} \left( c_{i,\sigma}^\dagger c_{i+1,\sigma} + \text{H.c.} \right) + U \sum_i n_{i\uparrow} n_{i\downarrow} - \mu \sum_{i,\sigma} n_{i\sigma} \tag{138}$$

These operators belong to the `lattice_model` object defined on line 4. Note that the chemical potential $\mu$ is added to the model automatically. Note also that the library only allows one `lattice_model` object to be defined at a time (even though its parameters may vary at will); it is possible to redefine (or reset) the lattice model and all cluster objects by calling the function `pyqcm.reset_model()`.

We next consider the Hubbard model on the honeycomb lattice, with the clusters illustrated on Fig. 15, and defined with the code below.

```
1  import pyqcm
2  import numpy as np
3  CM = pyqcm.cluster_model(4)
4  clus1 = pyqcm.cluster(CM, ((-1,-1,0), (0,1,0), (1,0,0), (0,0,0)), (1,0,0))
5  clus2 = pyqcm.cluster(CM, ((1,1,0), (0,-1,0), (-1,0,0), (0,0,0)), (-1,0,0))
6  model = pyqcm.lattice_model('graphene_4_2C', (clus1, clus2), ((4,2,0), (2,-2,0)),
       ((1,-1,0), (2,1,0)))
7  sq3 = np.sqrt(3.0)/2
8  model.set_basis(((1,0,0),(-0.5,sq3,0)))
```

```
9   model.interaction_operator('U')
10  model.hopping_operator('t', (-1,0,0), 1, orbitals=(1,2))
11  model.hopping_operator('t', (0,-1,0), 1, orbitals=(1,2))
12  model.hopping_operator('t', (1,1,0), 1, orbitals=(1,2))
```

In this case, the repeated unit contains two four-site clusters, the second being the inverted image of the first (note how the sites of each cluster are labeled on lines 4 and 5). The two clusters are defined with the same cluster model object `CM`, meaning that they will lead to different impurity problems based on the same Hilbert space and operators, possibly with different values of the terms in the Hamiltonian. Note that the call to the constructor `pyqcm.cluster()` contains a third, optional argument which is the base position of the cluster (here `(1,0,0)` and `(-1,0,0)`), added to the positions listed in the second argument. If the two clusters are expected to have the same Hamiltonian, one may avoid solving the second one by defining it in terms of the first, i.e., by issuing the function

```
1   clus2 = pyqcm.cluster(clus1, ((1,1,0), (0,-1,0), (-1,0,0), (0,0,0)), (-1,0,0))
```

instead, where the first argument is a `cluster` object instead of a `cluster_model` object. Each cluster inserted in the lattice model is given an index (from 0 to the number of clusters $-1$) in the order in which they are given to the function `pyqcm.lattice_model()`; this may be used later to query cluster specific information. Line 6 defines the lattice model, not only with the super-lattice vectors $(4, 2, 0)$ and $(2, -2, 0)$, but also with lattice vectors $(1, -1, 0)$ and $(2, 1, 0)$; the latter imply that the model contains two orbitals, associated with sub-lattices A and B. In `pyqcm`, each site-orbital pair lives on a distinct site of the lattice (like in graphene); if a model contains several orbitals on a given atom, then the `pyqcm` lattice is artificially given additional sites within the unit cell to incorporate these orbitals (this is in no way restrictive). The working basis is defined on line 7; this allows plotting routine to respect to geometry of the problem, but otherwise has no impact. Since we are dealing with a two-band model, the function calls on lines 9 to 11 that define the hopping terms must specify the initial and final orbitals of each hopping term, as well as the hopping direction.

## 8.3   Defining models II : Operators

In general, the lattice Hamiltonian is viewed as a sum of terms:

$$H = \sum_a h_a H_a \tag{139}$$

The various operators $H_a$ are defined by different functions depending on their types, as detailed below.

**One-body operators**   Operators of the type

$$H = \sum_{\mu\nu} t_{\mu\nu} c_\mu^\dagger c_\nu \tag{140}$$

where $\mu, \nu$ are composite indices comprising site, orbital and spin, can be defined with the `hopping_operator()` function. Each term in the above expression will be of the following form:

$$\sum_{ss'} \sum_{i,j} c_{is}^\dagger \tau_{ij}^{(a)} \sigma_{ss'}^{(b)} c_{js'} \tag{141}$$

where $i$ and $j$ run from 1 to 2 and correspond to the two sites of a pair, and $s, s'$ are spin indices (also from 1 to 2). The matrices $\tau^{(a)}$ $(a = 0, 1, 2, 3)$ and $\sigma^{(b)}$ $(b = 0, 1, 2, 3)$

are Pauli matrices (including the identity matrix). The above form guarantees that the expression is Hermitian, and the different possibilities for $a$ and $b$ correspond to different situations:

- $a = 1$ and $b = 0$ : A simple, spin-independent hopping term.
- $a = 2$ and $b = 0$ : A purely imaginary hopping term.
- $a = 0$ and $b = 3$ : A local Zeeman term in the $z$ direction.
- $a = 0$ and $b = 1$ : A local Zeeman term in the $x$ direction.
- $a = 1$ and $b = 1$ : A spin-flip hopping term, arising from a spin-orbit coupling.
- etc.

For instance, in the case of the graphene lattice above, an antiferromagnetic operator called `M` with opposite spins on the A and B sub-lattices could be defined as follows:

```
1  model.hopping_operator('M', (0,0,0), 1, orbitals=(1,1), tau=0, sigma=3)
2  model.hopping_operator('M', (0,0,0),-1, orbitals=(2,2), tau=0, sigma=3)
```

Note that different calls of the `hopping_operator()` function with the same operator name will just accumulate matrix elements for that operator.

**Interaction operators**   A Hubbard interaction of the form

$$H = \sum_i U_i n_{i\uparrow} n_{i\downarrow} \tag{142}$$

or an extended density-density interaction of the form

$$H = \sum_{ij} V_{ij} n_i n_j \qquad (n_i = n_{i\uparrow} + n_{i\downarrow}) \tag{143}$$

can be defined with the `interaction()` function. In the case of an extended interaction, the `link` argument, defining the relative position of the sites, must be provided. In multi-band models, the `orbitals` argument must also be specified, otherwise all possibilities are covered.

It is also possible to add a Hund coupling term:

$$H = \sum_{i,j} J_{ij} H_{ij} \tag{144}$$

where

$$H_{ij} = -n_{i\uparrow}n_{j\uparrow} - n_{i\downarrow}n_{j\downarrow} + c_{i\uparrow}^\dagger c_{j\uparrow} c_{j\downarrow}^\dagger c_{i\downarrow} + c_{j\uparrow}^\dagger c_{i\uparrow} c_{i\downarrow}^\dagger c_{j\downarrow} + c_{i\uparrow}^\dagger c_{j\uparrow} c_{i\downarrow}^\dagger c_{j\downarrow} + c_{j\uparrow}^\dagger c_{i\uparrow} c_{j\downarrow}^\dagger c_{i\downarrow} \tag{145}$$

This can also be written as

$$H_{ij} = -n_{i\uparrow}n_{j\uparrow} - n_{i\downarrow}n_{j\downarrow} + (c_{i\uparrow}^\dagger c_{j\uparrow} + \text{H.c.})(c_{i\downarrow}^\dagger c_{j\downarrow} + \text{H.c.}) \tag{146}$$

or as

$$H_{ij} = -c_{i\uparrow}^\dagger c_{j\uparrow}^\dagger c_{j\uparrow} c_{i\uparrow} - c_{i\downarrow}^\dagger c_{j\downarrow}^\dagger c_{j\downarrow} c_i \downarrow + c_{i\uparrow}^\dagger c_{i\downarrow}^\dagger c_{j\downarrow} c_{j\uparrow} + c_{j\uparrow}^\dagger c_{j\downarrow}^\dagger c_{i\downarrow} c_{i\uparrow} - c_{j\uparrow}^\dagger c_{i\downarrow}^\dagger c_{i\uparrow} c_{j\downarrow} - c_{i\uparrow}^\dagger c_{j\downarrow}^\dagger c_{j\uparrow} c_{i\downarrow} \tag{147}$$

In `pyqcm`, this is done by adding the argument `type='Hund'` to the function `interaction()`. Likewise, one may add a Heisenberg coupling

$$H = \sum_{i,j} J_{ij} \mathbf{S}_i \cdot \mathbf{S}_j \tag{148}$$

with the `type='Heisenberg'` option. Note however that `pyqcm` is designed for electron models, not spin models, meaning that the charge degree of freedom is always present. Therefore, this is not the most efficient tool to study pure quantum spin models.

**Anomalous operators**  When studying superconductivity, pairing operators must be defined:

$$H = \sum_{i,j,s,s'} \left( \Delta_{ij,b} c_{is} (i\sigma_b \sigma_2)_{ss'} c_{js'} + \text{H.c.} \right) \tag{149}$$

where the index $B$ can take the values 0 to 3. The case $b = 0$ corresponds to singlet superconductivity (in which case $\Delta_{ij,0} = \Delta_{ji,0}$) and the cases $b = 1, 2, 3$ corresponds to triplet superconductivity (in which case $\Delta_{ij,b} = -\Delta_{ji,b}$). In `pyqcm`, this is done via the function `anomalous_operator()`. For instance, in the case of the graphene lattice, an extended $s$-wave pairing (with equal amplitude on each bond) would be defined as

```
1  model.anomalous_operator('xS', (-1,0,0), 1, orbitals=(1,2), type = 'singlet')
2  model.anomalous_operator('xS', (0,-1,0), 1, orbitals=(1,2), type = 'singlet')
3  model.anomalous_operator('xS', (1,1,0), 1, orbitals=(1,2), type = 'singlet')
```

Other possible values of `type` would be `z`, `x` and `y`, for the possible directions of the **d**-vector describing triplet superconductivity.

**density waves**  Density wave operators are defined with a spatial modulation characterized by a wave vector **Q**. They can be based on sites or on bonds. If the operator is a site density wave, its expression is

$$x \sum_{\mathbf{r}} A_{\mathbf{r}} \cos(\mathbf{Q} \cdot \mathbf{r} + \phi) \tag{150}$$

where $A_{\mathbf{r}} = n_{\mathbf{r}}$ or $S_{\mathbf{r}}^z$ or $S_{\mathbf{r}}^x$. If it is a bond density wave, its expression is

$$\sum_{\mathbf{r}} \left[ x c_{\mathbf{r}}^\dagger c_{\mathbf{r+e}} e^{i(\mathbf{Q} \cdot \mathbf{r} + \phi)} + \text{H.c} \right] \tag{151}$$

where **e** is the bond vector. If it is a pair density wave, its expression is

$$\sum_{\mathbf{r}} \left[ x c_{\mathbf{r}} c_{\mathbf{r+e}} e^{i(\mathbf{Q} \cdot \mathbf{r} + \phi)} + \text{H.c} \right] \tag{152}$$

**e** is the link vector and **r** a site of the lattice. In `pyqcm` the different types of density waves are defined with the function `density_wave()` and different values of the argument `type` specify the type of density wave: `N` for a charge density wave, `Z` and `X` for spin density waves in the direction $z$ and $x$, `singlet` for a singlet pair density wave and `x`, `y` and `z` for pair density wave with triplet pairing and **d**-vector in the directions $x$, $y$ or $z$.

The wave-vector **Q** is given in argument to the function `density_wave()`, in multiples of $\pi$; for instance, Néel antiferromagnetism on a square lattice is specified as $\mathbf{Q} = (1, 1, 0)$. The function call in that specific example would be

```
1  model.density_wave('M', 'Z', (1,1,0))
```

(the first argument is the name given to the operator). Density wave operators must be commensurate with the repeated unit (super unit cell), but they can span several clusters if the latter is made of several clusters. Different local operators are then created on the clusters making up the repeated unit.[1]

---

[1]For internal reasons, these operators are given different names, so that different clusters based on the same `cluster_model` object are associated with the correct Hilbert space operator; these names are obtained by appending the string `@n` to the name of the lattice operator, where `n` is the cluster index (starting at 1). For instance, if the model is based on two clusters, the local implementation of the lattice density wave named `M` will be called `M@1` and `M@2`. These names are mostly for internal use and are not needed when specifying values (we can still use `M_1`, `M_2`, etc.), but the occasionally creep up in functions like `susceptibilty()` and `susceptibilty_poles()`.

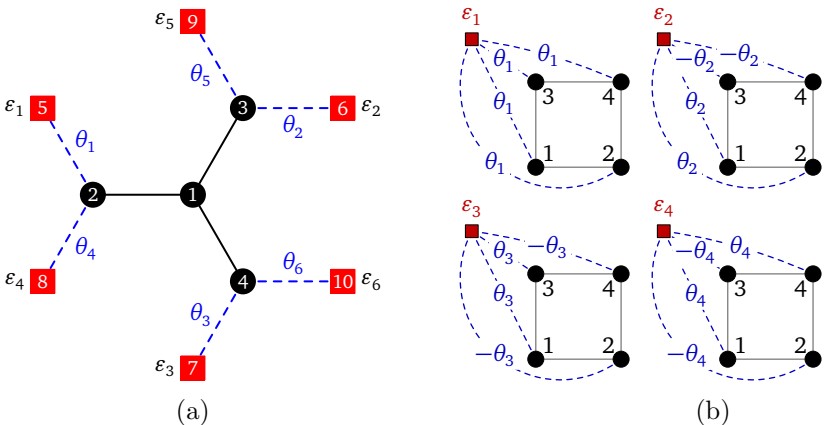

Figure 16

Note that if a model requires the definition of several clusters, then a different object of type `cluster_model` must be defined for each cluster containing different operators, except for density-waves. For instance, when modelling the three-band Hubbard model applied to the high-$T_c$ cuprates, one could define a cluster for, say, four copper atoms, another one for four oxygen atoms, and a third one, equivalent to the second, for another set of four oxygen atoms. Despite the fact that all three cluster have four sites and could be associated with similar bath configurations in CDMFT, two different cluster models must be defined: one for the first (copper) cluster, and one for the other two (oxygen) clusters. This is because the operators pertaining to the copper and oxygen clusters are different, whereas the two oxygen clusters have the same set of operators.

## 8.4 Cluster specific operators

The various operators defined on the lattice are used to define their restriction on the clusters making up the repeated unit. Thus there is no need to separately define operators on clusters, unless these operators have no equivalent on the lattice. This is the case of bath operators as used in CDMFT. Let us consider, for instance, a set of 6 bath sites added to the 4-site cluster of Fig. 15. Even though bath sites have no position, it is convenient in this case to represent them as in Fig. 16, by red squares. In `pyqcm` the degrees of freedom are numbered as follows: First the spin up operators $c_{i\uparrow}$ with $i = 1, \ldots N_s, N_s+1, \ldots, N_s+N_b$ where $N_s$ is the number of physical sites and $N_b$ the number of bath sites. Then the spin down operators $c_{i\downarrow}$, in the same order. The sites are thus labelled as illustrated in Fig. 16: physical sites first, followed by bath sites.

The various operators involving bath operators are then defined explicitly, by enumerating their matrix elements. For instance, the energy level $\varepsilon_1$ associated with the first bath site (labelled 5) on Fig. 16 would be defined as follows in an object `CM` of type `cluster_model`:

```
1  CM.new_operator('e1', 'one-body', ((5,5,1.0), (5+10,5+10,1.0)))
```

in which a list of two matrix elements is provided, each of the form $(i, j, v)$ with the indices $(i, j)$ of the degrees of freedom and the numerical value $v$. The hybridization operator noted $\theta_1$ in the figure would be defined as

```
1  CM.new_operator('theta1', 'one-body', ((2,5,1.0), (2+10,5+10,1.0)))
```

Note that the spin down part of the operator is represented by a matrix elements with spin down labels (obtained by adding $N_s + N_b = 10$ to the spin up labels).

## 8.5   Model instances and exact diagonalization

The values $h_a$ of the model parameters (see Eq. (139)) define an *instance* of the lattice model, and an unlimited number of such instances can be defined, either successively or concurrently (although they are usually defined in succession).

Even though operators are defined on the clusters from their definition on the lattice, the values of these operators (i.e. the values of the coefficients $h_a$) do not have to be the same on the lattice as on the clusters. Indeed, this is important in VCA, where the reference system (the cluster) has a non-interacting Hamiltonian that differs from the lattice Hamiltonian. For instance, the Néel antiferromagnetism operator M defined above for the model of Fig. 15 would, in the context of VCA, be zero on the lattice, but would serve as a Weiss field on the clusters. In pyqcm, the symbol associated with an operator (here M) is used to label the operator $H_a$, as well as its coefficient $h_a$. The value of the operator on cluster 1 would be labelled M_1, that on cluster 2 would be labelled M_2, etc. Since bath operators are only defined on clusters, their value is also labelled using an underscore followed by the cluster index, for instance eb1_1 and theta1_1 for the bath operators defined above (for this reasons, underscores cannot be used in operator names).

The values of the various operators must first be declared prior to building the first model instance, typically by the function lattice_model.set_parameters(), which takes a long string as argument, for instance like this:

```
1  model.set_parameters("""
2  U = 6
3  t = 1
4  mu = 3
5  M = 0
6  M_1 = 0.1
7  """)
```

Only the operators whose values have been declared like this will be effectively constructed in the Hilbert space of each cluster. Others will be ignored, even though they have been introduced earlier when defining the model.

By default, the values of the parameters on the clusters are inherited from that of the lattice Hamiltonian. Only when their value is explicitly specified (like M_1 above) are they different. It also possible to link the values of some parameters to others in order to obey some constraints; for instance the chemical potential could be set to $\mu = U/2$ by replace the line mu =3 above by mu = 0.5*U (only multiplications are allowed). This inheritance of values will be preserved even if the value of U is changed later. Once a parameter is declared dependent on another, it cannot regain its independence.

It is also important, before creating the first instance of the model, to specify in which Hilbert space sector of each cluster to look for the ground state. For instance, if the model conserves the number of particles and the $z$-component of the spin, the string R0:N4:S0 means that the ground state of the cluster must be searched for in the Hilbert space sector with $N = 4$ electrons and $S = 0$ spin projection. R0 means that the ground state presumably belongs to the trivial representation (labeled 0) of the point group (see the section on symmetries below). The spin projection is expressed by an integer $S = 2S_z$. Thus, S1 means $S_z = \frac{1}{2}$ and S-2 means $S_z = -1$. In the model illustrated in Fig. 15, one would need a statement like

```
1  model.set_target_sectors(['R0:N4:S0', 'R0:N4:S0'])
```

before defining the first instance of the model.

Hilbert space sectors are a crucial element of the use of the library and may be the source of physical errors. Performance issues dictate that not all Hilbert space sectors

should be checked for the true ground state for every calculation. Some judgement must be applied as to which sector or subset of sectors contains the true ground state. For a given cluster, a subset of sectors may be provided instead of a single one, by separating the sector keywords by slashes (/). For instance, the string indicating that the ground state should be searched in the sectors of the trivial representation, with N=3 electrons and spin projection $-1/2$ or $1/2$ is `R0:N3:S-1/R0:N3:S1`.

If spin is not conserved because of the presence of spin-flip terms, then the spin label must be omitted. For instance, the string `R0:N4` denotes the sector containing 4 electrons, in the trivial point group representation. An error message will be issued if the user specifies a spin sector in such cases, or inversely if the spin sector is not specified when spin is conserved.

The same is true in cases where particle number is not conserved, i.e., when pairing operators are nonzero: the number label must be omitted. For instance, the string `R0:S0` denotes the sector with zero spin, in the trivial point-group representation and an undetermined number of electrons.

When the target Hilbert space sector (or subset of sectors) specified by `lattice_model.set_target_sectors()` does not contain the true ground state, then the Green function computed thereafter will be wrong, because excited states obtained from the pseudo ground state by applying creation or annihilation operators may have a lower energy.

Once the active parameters have been declared and the target Hilbert space sectors specified for the `lattice_model` object `model`, one may call

```
I = pyqcm.model_instance(model)
```

to defined an object `I` of type `model_instance` that contains an instance of the model. By itself this does nothing, as the `pyqcm` library is "lazy" and will only work when specifically asked to, for instance by requesting the ground state properties of clusters of any quantity that involves the Green function. Internally (in the `qcm.so` library), model instances are labeled by integers to differentiate them. This is hidden in the Python interface as `model_instance` objects are created and one does not need to worry about it.

## 8.6    Green functions and CPT features

Once a model instance object `I` has been defined, the cluster's Green function can be accessed by the function `I.cluster_Green_function(z, clus)` where `clus`, is the cluster index (starts at 0, not 1) within the repeated unit and $z$ is a complex frequency. This function will return the Green function matrix for that frequency and cluster. In the absence of spin-flip or pairing terms, this matrix will be $L \times L$ ($L$ being the number of sites in the cluster). If the model is spin-dependent or if the ground state sector does not have zero spin projection, then the Green function will not be the same for up and down spins ($\mathbf{G}_\uparrow \neq \mathbf{G}_\downarrow$) and this function will return the spin up component; the spin down component can be obtained by adding the optional argument `spin_down=True`.

If spin is not conserved, then the returned Green function is $2L \times 2L$ and contains both spin-diagonal and spin-off-diagonal components. If particle number is not conserved, but spin is, then a restricted Nambu formalism is used and the Green function is also a $2L \times 2L$ matrix, this time containing both normal and anomalous components in terms of the Nambu spinor

$$\Psi = \left(c_{1,\uparrow}, \ldots, c_{L,\uparrow}, c_{1,\downarrow}^\dagger, \ldots, c_{L,\downarrow}^\dagger\right) \tag{153}$$

If neither spin nor particle number is conserved, then the Green function is a $4L \times 4L$ matrix in terms of the full, $4L$-component Nambu spinor

$$\Psi = \left(c_{1,\uparrow}, \ldots, c_{L,\downarrow}, c_{1,\uparrow}^\dagger, \ldots, c_{L,\downarrow}^\dagger\right) \tag{154}$$

The CPT Green function (55) is provided by the function `I.CPT_Green_function(z, k)`, where $z$ is a complex-valued frequency and `k` a wave-vector, specified by three components, in multiples of $2\pi$. For instance, `I.CPT_Green_function(1+0.05j, (0.5,0,0))` would return the CPT Green function at $z = \omega + i\eta = 1 + 0.1i$ and wave-vector $\mathbf{k} = (\pi, 0, 0)$. The CPT Green function has the same dimension as the cluster Green function if there is a single cluster in the repeated unit. Otherwise, its dimension is the sum of dimensions of the Green functions of the different clusters within the repeated unit and the indices pertaining to the different clusters appear in succession (i.e. the spin or Nambu indices, if any, of the first cluster, appear first, followed by those of the second cluster, and so on).

The periodized Green function (58) is provided by the function `I.periodized_Green_function(z, k)`, and returns a lower-dimensional matrix. If spin and particle number are conserved, its dimension is $N_b \times N_b$, where $N_b$ is the number of bands (or orbitals, as this is computed in the orbital basis). Again, if spin and/or particle number is not conserved, this is multiplied by 2 or 4. If one prefers the band basis, then the function `I.band_Green_function()` can be used instead, but its relevance in the presence of interactions is not clear.

The CPT Green function can be used to compute lattice averages of operators (see Eq. (52)). This is accomplished by the function `I.averages()` and the results are automatically appended to the file `averages.tsv`. This file also contains data on the ground state properties, like the wave-function average and variance of each operator on each cluster of the repeated unit.

The library contains various functions producing plots of spectral properties based on either the cluster Green functions or the CPT Green function. For instance, the function `I.spectral_function()` draws the spectral weight $A(\mathbf{k}, \omega)$ along a certain wave-vector path in a specified frequency domain. It can also draw the self-energy. The function `I.mdc()` draws a color plot of the spectral function in a plane of the Brillouin zone at a give frequency. The function `I.plot_DoS()` plots the local density of states (by integrating the CPT Green function over momentum) on a given frequency grid. The function `I.plot_dispersion()` plots the non-interacting dispersion relation, etc.

## 8.7 CDMFT

The submodule `pyqcm.cdmft` manages CDMFT computations. Its main component is the class constructor `CDMFT()`, which has a rather long list of parameters, most of them having default values. The first and only non-optional argument is the list of bath parameters used in the CDMFT procedure (these are generally called *variational parameters* in `pyqcm`). Let us give below a complete example of CDMFT usage, including the model definition, appropriate for the one-dimensional Hubbard model with a 4-site cluster and 4-site bath, as illustrated on Fig. 11(a):

```python
import pyqcm
CM = pyqcm.cluster_model(4, n_bath=4)
CM.new_operator('eb1','one-body',[(5,5,1.0),(6,6,1.0),(13,13,1.0),(14,14,1.0)])
CM.new_operator('eb2','one-body',[(7,7,1.0),(8,8,1.0),(15,15,1.0),(16,16,1.0)])
CM.new_operator('tb1','one-body',[(1,5,-1.0),(4,6,-1.0),(9,13,-1.0),(12,14,-1.0)
    ])
CM.new_operator('tb2','one-body',[(1,7,-1.0),(4,8,-1.0),(9,15,-1.0),(12,16,-1.0)
    ])
clus = pyqcm.cluster(CM, ((0,0,0), (1,0,0), (2,0,0), (3,0,0)))
model = pyqcm.lattice_model('1D_4_4b', clus, ((4,0,0),))
model.interaction_operator('U')
model.hopping_operator('t', (1,0,0), -1)

model.set_target_sectors(['R0:N8:S0'])
```

```
13  model.set_parameters("""
14  t=1
15  U=4
16  mu=2
17  eb1_1 = 1
18  eb2_1 =-1
19  tb1_1 = 1
20  tb2_1 = 1
21  """)
22
23  import pyqcm.cdmft as cdmft
24  solution = cdmft.CDMFT(model, varia = ('eb1_1', 'eb2_1', 'tb1_1', 'tb2_1'))
```

Lines 3–6 define the four bath operators (two bath energy levels `eb1` and `ebd2` and two hybridizations operators `tb1` and `tb2`). Line 7 defines the cluster with positions $(i, 0, 0)$ $(i = 0, 1, 2, 3)$, added in Line 8 to the repeated unit with super-lattice vector $(4, 0, 0)$. Lattice operators are defined on lines 9 and 10. Line 12 defines the expected ground state sector near or at half-filling (8 electrons, as we suspect the bath will be half-filled as well). Lines 13-21 set the initial values of the parameters, including bath parameters (note the suffix `_1`). Line 24 runs the CDMFT procedure per se. Progress appears on the screen. Each CDMFT iteration is recorded in a line appended to the file `cdmft_iter.tsv` and the converged solution is appended to the file `cdmft.tsv` (this file name is the default of an optional argument). By default, the distance function weights $W_n$ of Eq. (129) are uniformly distributed amongst Matsubara frequencies associated with a fictitious temperature $T = 1/\beta = 1/50$, up to a maximum $\omega_n = 2$, but these parameters are represented by the arguments `beta` and `wc` of the function `cdmft()` (see complete documentation for more details).

## 8.8   VCA

In the example below we reproduce the model used to generate Fig. 8, as well as the computation of the Potthoff functional for the $d_{x^2-y^2}$ symmetry. The cluster is a $2 \times 2$ plaquette.

```
1   import pyqcm
2   import numpy as np
3   CM = pyqcm.cluster_model(4)
4   clus = pyqcm.cluster(CM, ((0,0,0), (1,0,0), (0,1,0), (1,1,0)))
5   model = pyqcm.lattice_model('2x2', clus, ((2,0,0), (0,2,0)))
6   model.interaction_operator('U')
7   model.hopping_operator('t', (1,0,0), -1)
8   model.hopping_operator('t', (0,1,0), -1)
9   model.anomalous_operator('S', ( 0,0,0), 1)
10  model.anomalous_operator('D', (1,0,0), 1)
11  model.anomalous_operator('D', (0,1,0),-1)
12  model.anomalous_operator('xS', (1,0,0), 1)
13  model.anomalous_operator('xS', (0,1,0), 1)
14  model.anomalous_operator('Dxy', ( 1,1,0), 1)
15  model.anomalous_operator('Dxy', (-1,1,0),-1)
16
17  model.set_target_sectors(['R0:S0'])
18  model.set_parameters("""
19  t=1
20  U=8
21  mu=1.2
22  D_1 = 0.1
```

```
23  """)
24
25  for d in np.arange(1e-9,0.31,0.01):
26      model.set_parameter('D_1', d)
27      I = pyqcm.model_instance(model)
28      I.Potthoff_functional(file='sef_D.tsv')
```

Lines 7 and 8 define the hopping term (in the $x$ and $y$ directions) whereas Lines 10 and 11 define the $d_{x^2-y^2}$ pairing operator D. The other pairing operators associated with $s$-wave (S), extended $s$-wave (xS) and $d_{xy}$ pairing (Dxy) are also defined, but not used, since only D_1 is declared nonzero in the parameter declaration section. The function `Potthoff_functional()` is used to compute the Potthoff functional (115) and the result printed in the file given as argument.

Now this code snippet does not perform the VCA itself, which is an optimization procedure. This is done with the function `vca()` of the same submodule. For instance, the following call

```
1  import pyqcm.vca as vca
2  solution = vca.VCA(model, varia='D_1', steps=0.01, accur=2e-4, max=10, accur_grad
        =1e-8, method='altNR')
```

would perform an optimization of the Potthoff functional with the lattice model `model`, as a function of the Weiss field D_1, looking for a saddle point using a variant of the Newton-Raphson method (altNR), with an initial value of D_1=0.1 (as per the parameters declaration statement) and an initial step of 0.01. The method is set to fail if the absolute value of the Weiss field D_1 exceeds `max=10`, and converges if at some point the value of D_1 stops changing by `accur` or the estimated absolute value of the gradient falls below `accur_grad`.

Of course, the VCA can be performed with an arbitrary number of Weiss fields concurrently. The Weiss field optimization may be done using a variety of methods, including methods that look for strict minima, or pre-defined combinations of minima and maxima (see full documentation).

## 8.9 Extended interactions and Hartree approximation

In the presence of extended interactions, as explained in Sect. 7, one must carry out further approximations, in particular the Hartree approximation applied to the inter-cluster part of the interaction. In pyqcm, this is accomplished as follows.

One must first define the appropriate eigen-operators $m_\mu$ of Eq. (135). This can be done with the help of an additional module `cdw.py`, included in the distribution but not part of the pyqcm module *per se*. In that module, one just needs to specify the super-lattice vectors and the extended interaction, and the different $m_\mu$'s are then printed on the screen. Remains then to define them properly in the lattice model.

Let us consider, for instance, the one-dimensional, one-band Hubbard model with a cluster of length 4 and a nearest-neighbor interaction $V$, the latter defined by

```
1  model.interaction_operator('V', link=(1,0,0))
```

The two eigen-operators we need to keep are

$$m_0 = (n_1 + n_4)/\sqrt{2} \qquad\qquad m_1 = (n_1 - n_4)/\sqrt{2} \qquad\qquad (155)$$

with eigenvalues $\pm V$ respectively. The first one takes care of the Hartree shift to the chemical potential and the second one kicks in when a period-2 charge density wave appears. These two operators may be defined as follows:

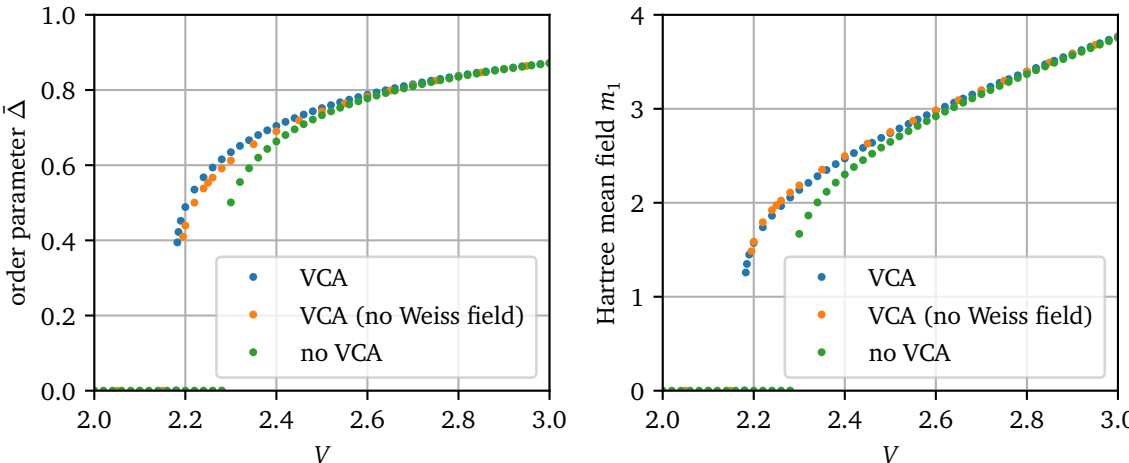

Figure 17: Left panel: CDW order parameter for the 1D extended Hubbard model at half-filling, as a function of the extended interaction $V$ with $U = 4$. The curved labeled VCA is obtained by treating the CDW Weiss field $\Delta'$ and the two Hartree fields $m_0$ and $m_1$ as variational parameters. The curved labeled 'no Weiss field' is obtained with $\Delta' = 0$. The curve labeled 'no VCA' is obtained simply by imposing the self-consistent condition on the Hartree mean fields $m_{0,1}$, using CPT to compute the average values. On the right panel, the Hartree mean field $m_1$ is shown, as a function of $V$.

```
1  e = np.sqrt(0.5)
2  model.explicit_operator('V0m', [((0,0,0), (0,0,0), e), ((3,0,0), (0,0,0), e)],
       tau=0, type='one-body')
3  model.explicit_operator('V1m', [((0,0,0), (0,0,0), e), ((3,0,0), (0,0,0),-e)],
       tau=0, type='one-body')
```

Then, the relation between these operators and the extended interaction $V$ must be encoded in objects of type `hartree`:

```
1  MF0 = pyqcm.hartree(model, 'V0m', 'V', 1, accur=0.01, lattice=True)
2  MF1 = pyqcm.hartree(model, 'V1m', 'V', -1, accur=0.001, lattice=True)
```

These couplings provide the names of the operators involved, the eigenvalues $D_\mu$ of Eq. (135) (here 1 and $-1$), the desired accuracy in the values of these parameters and the type of average value used for $\langle m_\mu \rangle$ (lattice or cluster average). Then, when performing the VCA or CDMFT, one may just add this list of couplings (the `hartree` argument of `VCA()` or `CDMFT`), or even perform a self-consistent Hartree procedure with only CPT, with the function `model.Hartree_procedure()`. In particular, the simple Hartree procedure is performed with the following function call:

```
1  def F(): return pyqcm.model_instance(model)
2  model.Hartree_procedure(task=F, couplings=(MF0,MF1), maxiter=256)
```

The VCA procedure could be obtained by the following calls:

```
1  def F():
2      V = model.parameters()['V']
3      model.set_parameter('mu', 2 + 2*V)
4      solution = VCA(model, varia=('cdw_1','V0m', 'V1m'), steps=0.001, hartree=(MF0
          ,MF1), hartree_self_consistent=False)
5      return solution.I
6  model.controlled_loop(task=F, varia=('cdw_1','V0m', 'V1m'), loop_param='V',
       loop_range=(3, 2, -0.02))
```

Note that the CDW order parameter is defined as

| name | default | meaning |
|---|---|---|
| continued_fraction | False | Uses a continued-fraction representation of the impurity Green function instead of a Lehmann representation. |
| print_Hamiltonian | False | Prints the many-body Hamiltonian matrix on the screen, if the dimension of the Hilbert space is small enough (see also parameter max_dim_print) |
| parallel_sectors | False | Distributes the different Hilbert space sectors (including those from point group symmetries) across different threads. |
| Davidson_states | 1 | Number of low-energy states to target for the ground state calculation. If 1, the Lanczos method is used to find the ground state. If $> 1$, the Davidson method is used. |
| max_iter_lanczos | 600 | Maximum number of iterations in the Lanczos method for the ground state. |
| accur_SEF | 5e-8 | Accuracy of the Potthoff functional computation. |
| temperature | 0 | Temperature used when targeting more than one low-energy state. This has to be low, since the Davidson method can only obtain a small number of low-energy states. Overall, the pyqcm solver remains an ED solver, not a finite temperature one. |
| Hamiltonian_format | 'S' | Format used to store or express the impurity Hamiltonian. 'S' means a compressed sparse-row (CSR) format. 'O' means that individual operators $H_a$ in the Hamiltonian are stored and applied in succession. 'F' means "factorized", and is possible when the Hamiltonian takes the form (73). 'N' means "None", in which case the action of the Hamiltonian is computed on the fly. 'E' means the the eigen library sparse matrix format is used, and is the generally the best option when available. |
| periodization | 'G' | Periodization scheme for the Green function. 'G' stands for the Green function scheme (58). 'M' stands for the cumulant periodization, 'S' for a periodization of the self-energy, etc. |

Table 2: A few of the global options of the pyqcm library.

```
1  model.density_wave('cdw', 'N', ( 1, 0, 0))
```

Fig. 17(a) shows the CDW order parameter as a function of $V$ for the one-dimensional extended Hubbard model, and Fig. 17(b) shows the corresponding values of the Hartree mean field $m_1$. There is little difference between including or not the CDW Weiss field $\Delta'$ in the procedure. However, performing the self-consistent procedure without the VCA with Hartree_procedure() yields slightly different results near the CDW transition, in particular a different value for the critical value $V_c$ of $V$ for the onset of charge order.

## 8.10   Global options

The pyqcm library contains a certains number of parameters with global effects, all listed in the documentation. These are set by the function set_global_parameter(<name>, <value>) and can be either boolean, integer, floating point values or chars. A few important examples are given in Table 2.

| Hamiltonian_format | OMP_NUM_THREADS | G.S. (s) | G.S. sym. (s) | G.F. sym (s) |
|:---:|:---:|:---:|:---:|:---:|
| S | 1 | 78 | 43.6 | 257 |
| S | 8 | 67.7 | 42.1 | 114 |
| E | 1 | 62.0 | 39.6 | 238 |
| E | 2 | 44.9 | 31.2 | 126 |
| E | 4 | 38.3 | 28.1 | 88 |
| E | 8 | 37.5 | 27.4 | 86 |

Table 3: Wall time (in seconds) for the ED processes on a linear chain of 14 sites at half-filling. The dimension of the Hilbert space is 11 778 624 in the $S_z = 0$, $N = 14$ sector (without symmetry). OMP_NUM_THREADS is the number of openMP threads. Computations are done on a M2 max processor under MacOS. Column 3 (G.S.) is the time to compute the ground state. Column 4 (G.S. sym) is the same, when taking the left-right mirror symmetry of the chain into account (the dimension of the Hilbert space is 5 889 312). Column 5 (G.F. sym) is the time needed to compute the ground state and the Green function representation, again with left-right symmetry.

## 8.11   Performance

The `pyqcm` library has limited parallelization capabilities, within openMP and MPI, as explained in this subsection. Different processes can be parallelized:

1. The matrix-vector product used in the various Lanczos methods

2. The construction of the Green function in the different symmetry sectors

3. The solution of the different clusters, if more than one

4. The frequency-momentum integrals

5. The simultaneous computation of the Potthoff functional at different points, in VCA

The `pyqcm` library is compiled against openMP. The number of threads is controlled, as usual, with the environment variable OMP_NUM_THREADS. Parallelism in openMP is used in many ways:

1. When constructing the Green function, different symmetry sectors (or the sectors with one more and one less electrons) are treated in parallel if the global option `parallel_sectors` is set.

2. When more than one cluster need to be solved, they are solved in parallel.

3. The matrix-vector product benefits from openMP if done with the `eigen` library (if the global parameter `Hamiltonian_format` is set to E).

Table 3 shows the computing (wall) time for the one-band Hubbard model on a chain of 14 sites for different number of threads (OMP_NUM_THREADS) on a M2 max processor, using both the in-house sparse matrix format for the Hamiltonian and the more efficient format from the `eigen`library. When computing the Green function, the global option `parallel_sectors` was set to true, which distributes the different band Lanczos procedures of Sect. 4.6 among the different threads. Be aware, however, that this increases the memory requirements considerably. If memory is not a problem, the rule of thumb is then to set

`OMP_NUM_THREADS` to twice the order of the symmetry group, e.g., 4 in the example studied in the table.

In the VCA procedure, several instances of the model need to be solved simultaneously, depending on the number of variational parameters and the optimization method used. In particular, the Newton-Raphson optimization method for the Potthoff functional requires $N_I = (n + 1)(n + 2)/2$ instances of the model to be solved per iteration, $n$ being the number of VCA variational parameters. The quasi-newton method (`SYMR1` or `BFGS`) requires $N_I = 2n + 1$ instances (it scales better as $n$ increases). These $N_I$ instances can be distributed over different computing nodes with MPI with almost perfect scaling (it is limited by the longest instance to be solved). MPI is used here at the Python level only (`mpi4py`) and issuing the command `mpirun -np <n_nodes> python <file.py>` suffices to exploit it.

**Acknowledgments** The authors are grateful to the numerous people who have used previous versions of this code over the years, or who have discussed some of the issues it faced. An incomplete list includes: S. Acheche, B. Bacq-Labreuil, M. Bélanger, M. Charlebois, S.S. Dash, J.P.L. Faye, O. Kaba, S. Kundu, X. Lu, A. Nevidomskyy, B. Pahlevanzadeh, P. Rosenberg, P. Sahebsara, A.-M. Tremblay and S. Verret.

**Funding information** DS acknowledges support by the Natural Sciences and Engineering Research Council of Canada (NSERC) under grant RGPIN-2020-05060. Computational resources were provided by the Digital Alliance of Canada and Calcul Québec.

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
