# Peer review of "Pyqcm: An open-source Python library for quantum cluster methods"

_SciPost Physics Codebases_

## Round 1 · Referee Report · Anonymous (Referee 1) · 2023-7-25

Report

The authors of the Python package "Pyqcm" describe in the manuscript the theoretical foundations and the usage of their uploaded code. The package has a Python interface and uses C++ functions. It enables the user to use cluster perturbation theory (CPT), variational cluster approximation (VCA), and cluster dynamical mean-field theory (CMDFT). The impurity solver is exact diagonalization.

First, I could install and use the package for the example calculations on my PC. Everything worked very smoothly on an Ubuntu distribution.
Second, I think the manuscript is very well written and presents many details about the theories' foundations and the code package's usage.

I support the publication of this manuscript in SciPost.
I have only a few suggestions and comments.

(1) While I could install the code on an Ubuntu distribution, I could not install it on a Mac with an Apple M1 Max chip. I got the same error message for the pip installation and a manual installation which read "Undefined symbols for architecture arm64."

(2) I think it would be helpful to include the position vectors in Fig. 1(a).

(3) It would be helpful for the reader to include a discussion on the limitations of the methods. The reader should be able to understand when CPT, VCA, and CDMFT can be used and when not.

---

## Round 1 · Referee Report · Anonymous (Referee 2) · 2023-8-22

Report

In their manuscript, the authors present an open-source Python library -based on C++ routines- for the quantum cluster methods cluster perturbation theory (CPT), variational cluster approximation (VCA) and cluster dynamical mean-field theory (CDMFT) using an exact diagonalization (ED) solver. The paper can be roughly divided into two parts, a first theory part where the quantum cluster methods are introduced, and a second part which concerns the library itself.

The manuscript is well structured, clearly written and I think that it will serve as an important read for novel users of these cluster methods. It is very complete, mentions the most common applications of VCA and CDMFT and also discusses some numerical details that one might encounter in practise when using the pyqcm package.
I strongly recommmend its publication in SciPost after some minor corrections in part one and some installation-related issues of the code have been taken care of.

Comments:
(1) The theory section is largely based on very instructive lecture notes of the corresponding author (arXiv:0806.2690). Whereas the arXiv preprint has to the best of my knowledge not been published before, some excerpts of it -including Fig.1- seem to be part of a Springer book from 2012 ('Strongly Correlated Systems') and might need to be properly referenced.

(2) I did not succeed to install the code on a Mac with Intel Core i7, neither on our local compute cluster or a large-scale compute facility. In both cases the openMP parallelization seems to cause problems, in particular when using Intel compilers with an up-to-date openMP version (v5.0, apparently related to nested critical regions). I finally managed to install it on a linux system using a gnu-compiler, but I think that the installation on Mac systems and/or Intel compilers with openMP would merit some additional guidance in the INSTALL file.

Some suggestions for further improvement of the manuscript:
(3a) In section 3.3 both the G- and M-periodization schemes are introduced. Could the authors give some guidance when to use the respective schemes? Also, what is the authors' opinion on the self-energy periodization scheme outlined in PRB 105, 245115 (2022) - could it avoid the issues mentioned in the manuscript and render the S-periodization a valid alternative?
(3b) When deriving the self-energy functional theory in section 5.1, the authors suppose that Eq.(102) can be locally inverted. This might be a detail, but perhaps the authors want to briefly mention the discussion about the (non-)invertability of this equation, see e.g. PRL 114, 156402 (2015) and arXiv:1407.6599.
(3c) The authors might want to mention some limitations of the techniques, for instance jumps in the self-energy functional when tuning a symmetry-breaking Weiss field within VCA, and how to deal with them.
(3d) In section 5.7 examples for minima and maxima of the SEF w.r.t. Weiss fields are given. Are there examples for true saddle points of the SEF w.r.t to a Weiss field, perhaps in the context of spin-orbit coupling, or are saddle points not of interest for practical purposes?

Typos etc.:
(4a) p.3 'one-site' -> 'on-site'
(4b) p.4 'labeled by vector' -> 'labeled by vectors'
(4c) p.7 'z' mentioned after Eq.(28) without being used there (only needed later in Eq. 29).
(4d) p.10 'space E one-electron states' -> 'space E of one-electron states'
(4e) p.13 1st sentence of last paragraph: This statement is true for half-filling, but not in general; simply take a sector with an odd number of electrons...
(4f) p.19 'with points groups' -> 'with point groups'
(4g) p.22 'over sites indices' -> 'over site indices'
(4h) p.23 Around Eq. 104 the commas and points after eqs. need to be checked
(4i) p.40 'function calls on lines 9 to 11' -> 'lines 10 to 12'
(4j) p.42 'index B' -> 'index b'

---

## Editorial Decision

resubmitted